# Solvent control of water O−H bonds for highly reversible zinc ion batteries

Yanyan Wang [1,7], Zhijie Wang [1,7], Wei Kong Pang [2], Wilford Lie [3],
Jodie A. Yuwono [1,4], Gemeng Liang [1], Sailin Liu [1], Anita M. D' Angelo [5],
Jiaojiao Deng [6], Yameng Fan [2], Kenneth Davey [1], Baohua Li [6] ✉ &
Zaiping Guo [1] ✉

Aqueous Zn-ion batteries have attracted increasing research interest; however, the development of these batteries has been hindered by several challenges, including dendrite growth, Zn corrosion, cathode material degradation, limited temperature adaptability and electrochemical stability window, which are associated with water activity and the solvation structure of electrolytes. Here we report that water activity is suppressed by increasing the electron density of the water protons through interactions with highly polar dimethylacetamide and trimethyl phosphate molecules. Meanwhile, the Zn corrosion in the hybrid electrolyte is mitigated, and the electrochemical stability window and the operating temperature of the electrolyte are extended. The dimethylacetamide alters the surface energy of Zn, guiding the (002) plane dominated deposition of Zn. Molecular dynamics simulation evidences $Zn^{2+}$ ions are solvated with fewer water molecules, resulting in lower lattice strain in the $NaV_3O_8 \cdot 1.5H_2O$ cathode during the insertion of hydrated $Zn^{2+}$ ions, boosting the lifespan of $Zn \| NaV_3O_8 \cdot 1.5H_2O$ cell to 3000 cycles.

The development of wind and solar energy sources needs suitable energy storage to integrate generated energy into the electricity grid[1]. Zn-ion batteries (ZIBs) are practically promising because of their safety, low cost, abundance of Zn and environmental 'friendliness' properties[2]. However, drawbacks of aqueous ZIBs include a narrow electrochemical stability window (ESW), corrosion of the Zn anode, dendrite growth, poor temperature adaptability, and dissolution of cathode materials[3]. Importantly, electrolytes have a significant impact on overall performance of ZIBs[4,5]. Aqueous electrolytes exhibit electrochemical decomposition and gas generation because of the narrow ESW for water, hindering application of high-voltage cathodes[6]. Because of the existence of $H_3O^+$ in mild-acid electrolytes, Zn corrosion occurs, causing low reversibility of the Zn anode[7]. Dendrite

growth[8], derived from uneven $Zn^{2+}$ deposition at the electrode/electrolyte interface, is highly significant in determining lifespan of ZIBs. Because of the freezing and boiling point for water, respectively, 0 and 100 °C, the majority of aqueous ZIBs do not work in harsh environments. In addition, the dissolution of cathodes is observed in aqueous electrolytes, which causes active material loss and rapid capacity fading[9].

To address this, various approaches have been reported including, constructing artificial protection layers, adding functional additives[10] and regulating $Zn^{2+}$ ion deposition behavior[11]. However, practical difficulty remains because these challenges arise from the solvent, water. It is widely acknowledged that the self-ionization reaction exists with pure water, in which $H_2O$ ionizes to form $OH^-$ and

¹School of Chemical Engineering & Advanced Materials, The University of Adelaide, Adelaide, SA 5005, Australia. ²Institute for Superconducting & Electronic Materials, University of Wollongong, Wollongong, NSW 2500, Australia. ³School of Chemistry and Molecular Bioscience, University of Wollongong, Wollongong, NSW 2500, Australia. ⁴College of Engineering and Computer Science, Australian National University, Canberra, ACT 2601, Australia. ⁵Australian Synchrotron, Australian Nuclear Science and Technology Organisation (ANSTO), Clayton, VIC 3168, Australia. ⁶Shenzhen Key Laboratory on Power Battery Safety and Shenzhen Geim Graphene Center, Tsinghua Shenzhen International Graduate School, Tsinghua University, Shenzhen 518055, China. ⁷These authors contributed equally: Yanyan Wang, Zhijie Wang. ✉e-mail: libh@mail.sz.tsinghua.edu.cn; zaiping.guo@adelaide.edu.au

$H_3O^+$ ($2H_2O \rightleftharpoons H_3O^+ + OH^-$). Together with the dissolution of zinc salts, such as $ZnSO_4$ and zinc trifluoromethanesulfonate ($Zn(OTf)_2$), the pH value reduces from near 7 for pure water, to ca. 5 for 1 M $ZnSO_4$/ $Zn(OTf)_2$ aqueous solution, because $Zn^{2+}$−water interaction causes water molecules to ionize. The electric field for $Zn^{2+}$ exerts a force on water molecules to induce electron transfer from coordinated $H_2O$ to empty orbitals of $Zn^{2+}$, which significantly weakens the O−H bonds of $H_2O$ molecules and promotes hydrogen evolution[12,13]. Theoretically, the self-ionization of water cannot be restrained by limited additives, therefore, practically, significant additive is necessary to ensure intensive interactions with water. Highly concentrated 'water-in-salt' electrolyte is an example in which dissolved salts far outnumber water, confining water molecules in ion solvation shells. Through suppressing $Zn^{2+}$−$H_2O$ interaction, hydrolysis of zinc salt is eliminated and the pH value for water-in-salt electrolytes approaches pH = 7[12]. However, this is not cost-effective and organic electrolytes are considered to eliminate $H_3O^+$. Organic electrolytes are advantageous in ESW, together with operational temperature range and thermodynamic stability with metallic Zn anode[14,15]. A drawback, however, is that most of these electrolytes are flammable and therefore are a safety risk in ZIBs. In addition, organic electrolytes have high charge-transfer impedance and a high desolvation penalty at the electrode/electrolyte interface, restraining the cathode from exhibiting full capacity[16]. To obviate these drawbacks in using aqueous and organic electrolytes, it was hypothesized that a judiciously designed hybrid electrolytes might be preferable[17]. Various non-aqueous solvents, such as dimethyl sulfoxide (DMSO)[18], dimethyl carbonate (DMC)[19], N-methylpyrrolidone (NMP)[20], triethyl phosphate (TEP)[21], diethyl carbonate (DEC)[22], propylene carbonate (PC)[23], and ethylene glycol (EG)[24], were added into aqueous electrolytes to suppress water decomposition[2]. The mechanism(s) is to break hydrogen bonds between water molecules, restrict activity of water in solvation sheaths, or exclude water molecules from the electric double layer (EDL).

In this work, we propose that water reactivity can be suppressed by increasing the electrostatic attraction between O−H via increasing the electron density of the water protons. Dimethylacetamide (DMAC) and trimethyl phosphate (TMP) are selected because they are highly polar molecules with electron-rich regions, C=O and P=O groups, that exert interaction on both solvated and free water molecules, and therefore impact the O−H bond strength of water. The decomposition of $H_2O$ in the hybrid electrolyte is less thermodynamically favorable compared with that in aqueous electrolyte. By confining the activity of $H_2O$, $H_2O$ decomposition-related hydrogen evolution, together with undesired side reactions are obviated. The Zn anode in the as-prepared electrolyte exhibits high plating/stripping efficiency of 99.5% over 2000 cycles at a current density of 1 mA cm$^{-2}$, and boosted anti-corrosion characteristics. DMAC alters the surface energy of Zn and therefore guide the (002) plane preferred orientation of Zn deposition, which boosts the lifespan of the Zn anode to >1600 h despite high applied current density and plating capacity of, respectively, 5 mA cm$^{-2}$ and 5 mAh cm$^{-2}$. Importantly, the $NaV_3O_8\cdot1.5H_2O$ (NVO) cathode exhibits significantly reduced lattice distortion during (de)intercalation of $Zn^{2+}$ in the electrolyte, boosting long-term cycling performance. The strong interactions between water and DMAC/TMP break the hydrogen-bonded water structure and reduce the dissociation of water at high temperature, enabling a wide operational temperature range of −40 to 70 °C for Zn∥NVO battery.

## Results

Electrolyte formulation was based on three principles, namely: (1) dendrite-free deposition of Zn; (2) good compatibility with NVO cathode, and (3) non-flammability. It was observed that the Zn deposition orientates the (002) plane when the volume percentage of water in DMAC/$H_2O$ mixture ranged from 20 to 40% (Supplementary Fig. S1). Water content has a highly significant impact on cathode

performance. The NVO electrode exhibited a satisfactory capacity and cyclic stability when the ratio of $H_2O$ was 30% (Supplementary Fig. S2). TMP is added to ensure the formulated electrolyte is nonflammable, and usage is ca. 30% in the mixture of DMAC/TMP (Supplementary Fig. S3). Therefore, the volume ratio for DMAC/TMP/$H_2O$ was determined to be 5:2:3. 1 M $Zn(OTf)_2$ dissolved in this mixture is denoted HE while 1 M $Zn(OTf)_2$ aqueous solution, AE.

In dilute aqueous electrolytes, including AE, $Zn^{2+}$ coordinates with six water molecules to form $Zn[H_2O]_6^{2+}$[12]. To determine the solvation structure for HE, molecular dynamics (MD) simulations were carried out. It is found that organic solvents, DMAC and TMP, together with the anion were evidenced to take part in the solvation shells (Supplementary Fig. S4). Radial distribution functions (RDFs) computed the distribution of the nearest-neighbor molecules around a reference $Zn^{2+}$, Fig. 1a. The coordination number for HE is six, which includes 3.7 $H_2O$ molecules, 1 DMAC molecule, 0.5 TMP molecule and 0.8 $OTf^-$ anion, based on the statistical findings. Fourier transform infrared (FT-IR) spectra that are significantly sentive to molecular dipole moment change, was applied to determine how polar molecules impact the O−H bonds of water. For AE, the absorbance bands in the region of 2800 to 3800 cm$^{-1}$ are related to O−H stretching vibrations. The second derivative spectra located the three main peaks, 3204, 3375 and 3540 cm$^{-1}$, corresponding to, respectively, network water, intermediate water and poorly connected water[25], Fig. 1b. It was found that these peaks all undergo apparent blueshift in the spectra for HE, demonstrating that hydrogen bonds are weakened while O−H bonds strengthened[23]. This finding is different from the previous report, that adding DMAC into the aqueous electrolyte leads to redshift of O−H vibrations because of increasing number of hydrogen bonds. The ratio of DMAC/$H_2O$ is the crucial factor that determines whether the hydrogen bonds are strengthened or weakened[26]. $^1$H nuclear magnetic resonance (NMR) spectroscopy confirmed that $H_2O$ molecules are under influence from other molecules in HE. As is shown in Fig. 1c, water protons resonate at ca. 3.7 ppm for pure water. This peak broadens in AE because water molecules are involved in the solvation clusters and induced by $Zn^{2+}$ accompanying a partial transfer of electron from the water proton to the empty orbitals of $Zn^{2+}$. The reduced electron density on the water proton deshields the H nucleus, and the proton resonance frequency shifts downfield. The water protons have a significant upfield shift of ~3.52 ppm in HE, evidencing that the electron density of water proton for HE increases compared with that for pure water and AE. Electron-rich regions of DMAC and TMP, C=O and P=O groups, increase the electron density of water protons when interacting with water, resulting in shielding of water protons. This significant shift of water proton resonance frequency highlights that the dipole-dipole interactions between DMAC/TMP and water are strong, and that it exists both in the solvation sheaths of $Zn^{2+}$ and amongst the molecules that are out of the solvation structure.

Under the influence of DMAC/TMP, water molecules exhibit boosted O−H bonds and become less likely to decompose. The deprotonation energy for $H^+$ dissociation from $H_2O$ in the $Zn^{2+}$ inner solvation shell was therefore investigated using density functional theory (DFT), Fig. 1d. For AE, the deprotonation energy for $[Zn(H_2O)_6]^{2+}$ was computed to be −4.94 eV. In HE, using the representative solvation structure $[Zn(H_2O)_3(DMAC)_1(TMP)_1(OTf)_1]^+$, the deprotonation energy is −2.43 eV on average, evidencing that $H^+$ dissociation from $H_2O$ is less favorable in HE than in AE. The ESW of HE was measured by linear sweep voltammetry (LSV), and it is stable up to 2.11 V vs. Ag/AgCl at a scanning rate of 1 mV S$^{-1}$, greater than that for AE (1.83 V), confirming the higher stability of water molecules in HE than that in AE (Fig. 1e). 3-electrode cyclic voltammetry (CV) was conducted to evaluate the reversibility of Zn in AE and CE, respectively. The Coulombic efficiency (CE) of Zn plating/stripping is 97.1% for HE, higher than 91.6% of AE (Supplementary Fig. S5). Figure 1f presents the lifespan and CE for Zn∥Cu batteries using AE and HE as the electrolyte

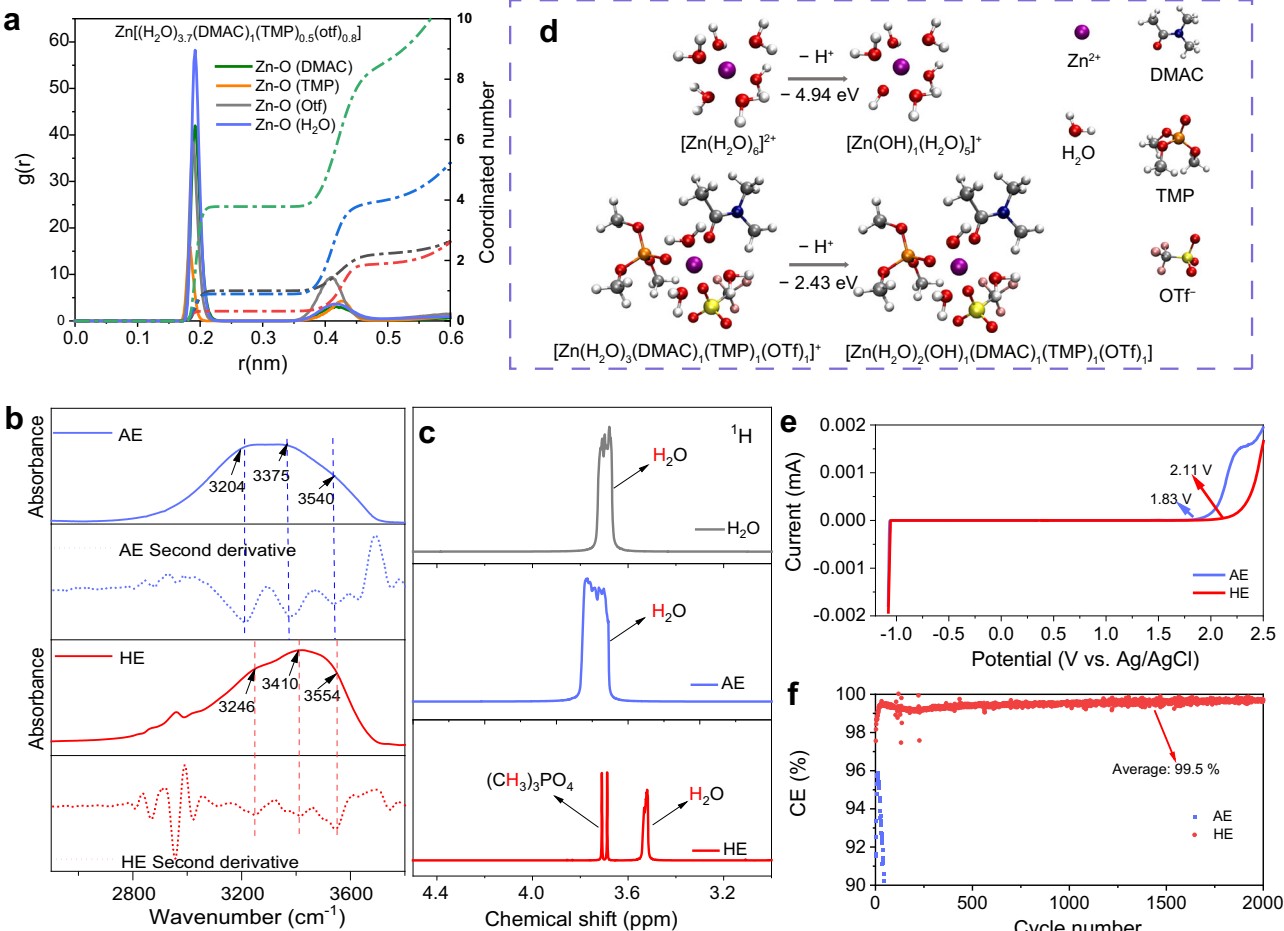

**Fig. 1 | Electrolyte structure and water O–H bond characterization. a** RDFs for $Zn^{2+}$–O pairs in HE. **b** FT-IR absorption and second derivative spectra for AE and HE. **c** $^1H$ NMR spectra for pure water, AE and HE. **d** Deprotonation (H+ dissociation of $H_2O$) energy from $Zn^{2+}$ inner solvation shell of AE and HE. **e** ESW for AE and HE measured with 10 μm Pt ultramicroelectrode as working electrode at a scan rate of 1 mV s⁻¹. **f** CE for Zn||Cu cells using AE and HE as the electrolyte, respectively. The current density is 1 mA cm⁻² and the plating capacity is 1 mAh cm⁻².

at, respectively, a current density 1 mA cm⁻² and plating/stripping capacity of 1 mAh cm⁻². It is seen in the figure that HE exhibits a highly significant improvement in stability and reversibility for the Zn anode with a CE as high as 99.5%.

Solvents significantly impact the morphology evolution of the formation of solid metallic Zn from the liquid electrolyte. When deposited in AE, 'flaky' metallic Zn loosely piles up, resulting in a porous deposition layer on the copper current collector, Fig. 2a. When this flaky Zn loses contact with the conductive substrate, it becomes inactive and irreversible in subsequent plating/stripping. Moreover, the non-planar, porous morphology leads to an inhomogeneous local charge density distribution and therefore induces unwanted dendrite growth that leads to a short-circuit. In contrast, the deposited Zn in HE exhibits a close-packed and stepped-surface with a hexagonal-like crystalline structure, Fig. 2b, a typical characteristic of (002) plane preferred crystal orientation[27–29]. In addition, intensity of the 002 reflection in its X-ray powder diffraction (XRD) pattern increases significantly compared with that in commercial Zn foil and Zn deposited in AE (Supplementary Fig. S6). The exposed (002) basal plane has a relatively smooth surface and therefore conducive to dendrite-free deposition of $Zn^{2+}$ ions. To more directly observe Zn electrodeposition, real-time images of the Zn/electrolyte interface were captured via an optical microscope, Fig. 2c. It can be seen in the figures that several 'sharp' protuberances appear in the initial stage when Zn is deposited in AE, and these continually grow because additional $Zn^{2+}$ is attracted and accumulated on the tips. In contrast, Zn deposition in HE is

significantly more uniform and compact, and no protuberance is seen during the entire deposition. Because a principal requirement for a Zn anode is to remain robust with extensive cycling, galvanostatic plating/stripping with Zn||Zn batteries is widely used as a test to determine the lifespan in different electrolytes. When a 'regular' current density of 1 mA cm⁻² and plating capacity 1 mAh cm⁻² was applied, the lifespan of the Zn anode in AE was ca. 750 h, while it was >3200 h in HE (Supplementary Fig. S7). When the current density and cycling capacity were increased to, respectively, 5 mA cm⁻² and 5 mAh cm⁻² as is presented in Fig. 2d, the Zn||Zn battery with AE as electrolyte has an apprent voltage decrease after 70 h, indicating the anode failed because dendrites penetrated the separator and caused a short-circuit. However, in HE the anode worked stably for >1600 h. This finding is attributed to the preferred orientation of the (002) plane during Zn plating, which has dendrite-free morphology during cycling.

Because the solidification transition for $Zn^{2+}$/Zn occurs in a solvent medium, we investigated how salt concentration and solvent components of the electrolyte impact morphology of deposited Zn. It was concluded that salt concentration was not the decisive factor because the special orientation appears also in 0.5, 1.5, and 2 M hybrid solution of Zn(OTf)₂ (Supplementary Fig. S8). With DMAC and TMP as the single solvent and 1 M Zn(OTf)₂ as salt, the morphology for Zn deposited in these two electrolytes is porous and irregular without preferred orientation (Supplementary Fig. S9). It was concluded that DMAC/TMP/H₂O hybrid solvents are crucial in Zn plating. Generally, crystallographic texture is determined by different growth rates

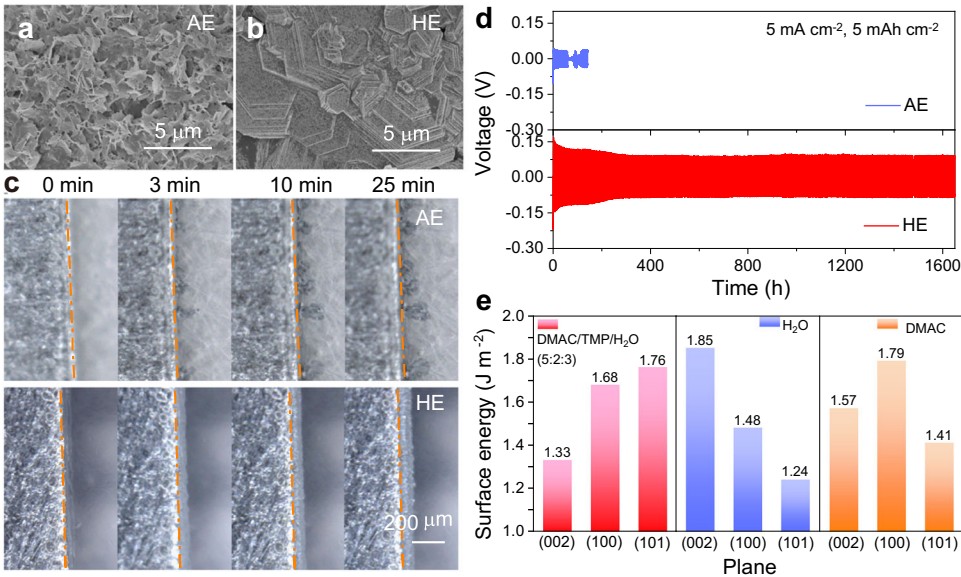

**Fig. 2 | Zn deposition in HE and AE.** Surface morphology for deposited Zn in **a** AE and **b** HE. **c** Optical microscopy image of Zn/electrolyte interface during Zn deposition in AE and HE. **d** Voltage profile for Zn∥Zn symmetric battery tested at, respectively, high current density and high deposition capacity with AE and HE as electrolyte. **e** Surface energy value for main planes of metallic Zn in the liquid environment of DMAC/TMP/$H_2O$ (5:2:3 by volume), $H_2O$ and DMAC, respectively.

amongst crystallites of various orientations, governed by surface energy minimization[30]. DFT computations were therefore carried out to determine the surface energy of three main planes, namely, Zn (002), (100), and (101), in a solvent environment. Surface energy, the excess energy presented at a particular surface, determines the exposed facets and growth orientation of the crystal. It can be modified by solvent molecules via interactions[31,32]. As a result, the surface energy values for Zn (002), (100), and (101) alter when they are exposed to different solvents. Supplementary Fig. S10 shows that Zn (002) exhibits the lowest surface energy of 1.33 J m$^{-2}$, in the mix of DMAC/TMP/$H_2O$ (5:2:3 by volume) compared with the (100) and (101) planes, which are, respectively, 1.68 and 1.76 J m$^{-2}$. This thermodynamic difference implies that the metallic Zn maximizes expression of the (002) plane to minimize the total surface energy of the crystal, accounting for the dominant (002) plane in metallic Zn growth. Figure 2e (and Supplementary Figs. S11, 12) compare the surface energy values of these planes in pure $H_2O$ and DMAC. It is seen that the (002) plane is no longer the thermodynamically preferred orientation in $H_2O$, while (100) and (101) planes have significantly lower values at, respectively, 1.48 and 1.24 J m$^{-2}$. In DMAC, the surface energy for the (101) plane is less than that for the (002) plane, and therefore (101) has the greatest proportion of exposure and not (002) plane.

The OTf$^-$ anion in the electrolyte also interacts with the solvent. To confirm the solvation structure for OTf$^-$ anion, Raman spectroscopy was carried out on Zn(OTf)$_2$ and solutions with various solvents, Fig. 3a. The −CF$_3$ symmetric and asymmetric deformation mode for Zn(OTf)$_2$ powder is seen at 767.9 cm$^{-1}$ and 584.5 cm$^{-1}$, respectively. It was found that these two peaks shift slightly in AE, evidencing an interaction between $H_2O$ and OTf$^-$ anion. The signals for −CF$_3$ shift when Zn(OTf)$_2$ is dissolved in TMP and DMAC, evidencing strongly that OTf$^-$ anion coordinates with them. However, the signals for −CF$_3$ overlap with that for TMP and DMAC molecules (Supplementary Fig. S13), making it practically difficult to identify which solvent molecule the OTf anion interacts with most within the HE. Nuclear magnetic resonance (NMR) spectroscopy can be used for complementary data on the chemical environment of the OTf$^-$ anion. Figure 2b shows the $^{19}$F resonance frequencies for Zn(OTf)$_2$ solutions with, respectively, $H_2O$, TMP, and DMAC as a single solvent. The $^{19}$F signal for these electrolytes appears as a 'sharp', narrow peak at ca.

−77.8 ppm. Because of different coordination molecules, the chemical shifts are different. The chemical shift for $^{19}$F in HE is the same as that in DMAC, evidencing that OTf$^-$ anions have the same chemical environment in DAMC solution and HE. In other words, OTf$^-$ anions are surrounded mainly by DMAC molecules in HE. In a single solvent, DMAC, $H_2O$, and TMP all coordinate with OTf$^-$ anion to some degree, however, competition for anions occurs in DMAC/$H_2O$/TMP mixture. DMAC molecule gains because its electron-rich acyl group exhibits much stronger interaction with −CF$_3$, a strong electron-withdrawing group. Consequently, the OTf$^-$ anion in HE is similar in character with the DMAC solution, with (almost) no anion decomposition.

Corrosion of the Zn anode was determined by immersing the Zn foil in AE and HE for 72 h. Zn foil exhibited severe corrosion in AE and was covered with a layer of anion-derived by-products (Supplementary Figs. S14b, d). However, Zn foil immersed in HE exhibits a smooth surface with only 'slight' corrosion spotting, which is confirmed by the smaller corrosion current (icorr) obtained in the Tafel plots (Supplementary Figs. S14c, e). A quantitative analysis of the corrosion rate of metallic Zn in these two electrolytes was conducted by monitoring the potential changes of Zn over time. Initially, a fixed amount of Zn, 0.5 mAh, was deposited onto the Ti foil to form a Zn@Ti electrode, and when all metallic Zn was corroded by the electrolyte, the potential of the Zn@Ti electrode increased significantly. It is calculated that the corrosion rate of Zn in AE is ca. 0.043 mg h$^{-1}$, while that in HE is significantly reduced to just 0.003 mg h$^{-1}$, (Supplementary Fig. S15). X-ray photoelectron spectroscopy (XPS) was used to analyze the surface chemistry of Zn foil that had been cycled 100 times in AE and HE. For the Zn foil cycled in AE, strong signals appeared in the S 2p and F 1s spectrum, evidencing side reaction products from electrolyte decomposition on Zn foil, specifically, ZnSO$_4$, ZnS, and ZnF$_2$[12], Fig. 3c. In contrast, no apparent by-products were detected on the Zn foil cycled in HE, Fig. 3d, confirming the side reactions between electrolyte and metallic Zn are significantly suppressed. The significantly improved stability of HE is attributed to the strong DMAC/TMP − $H_2O$ and DMAC−OTf$^-$ interaction that restricts activity of water and anion. Moreover, a preferred orientation for the Zn (002) plane is found for the Zn foil cycled 100 times in HE, confirming that the close-packed growth model with a layer-by-layer structure occurs in the initial plating and is maintained in the following cycling, Fig. 3f. In contrast,

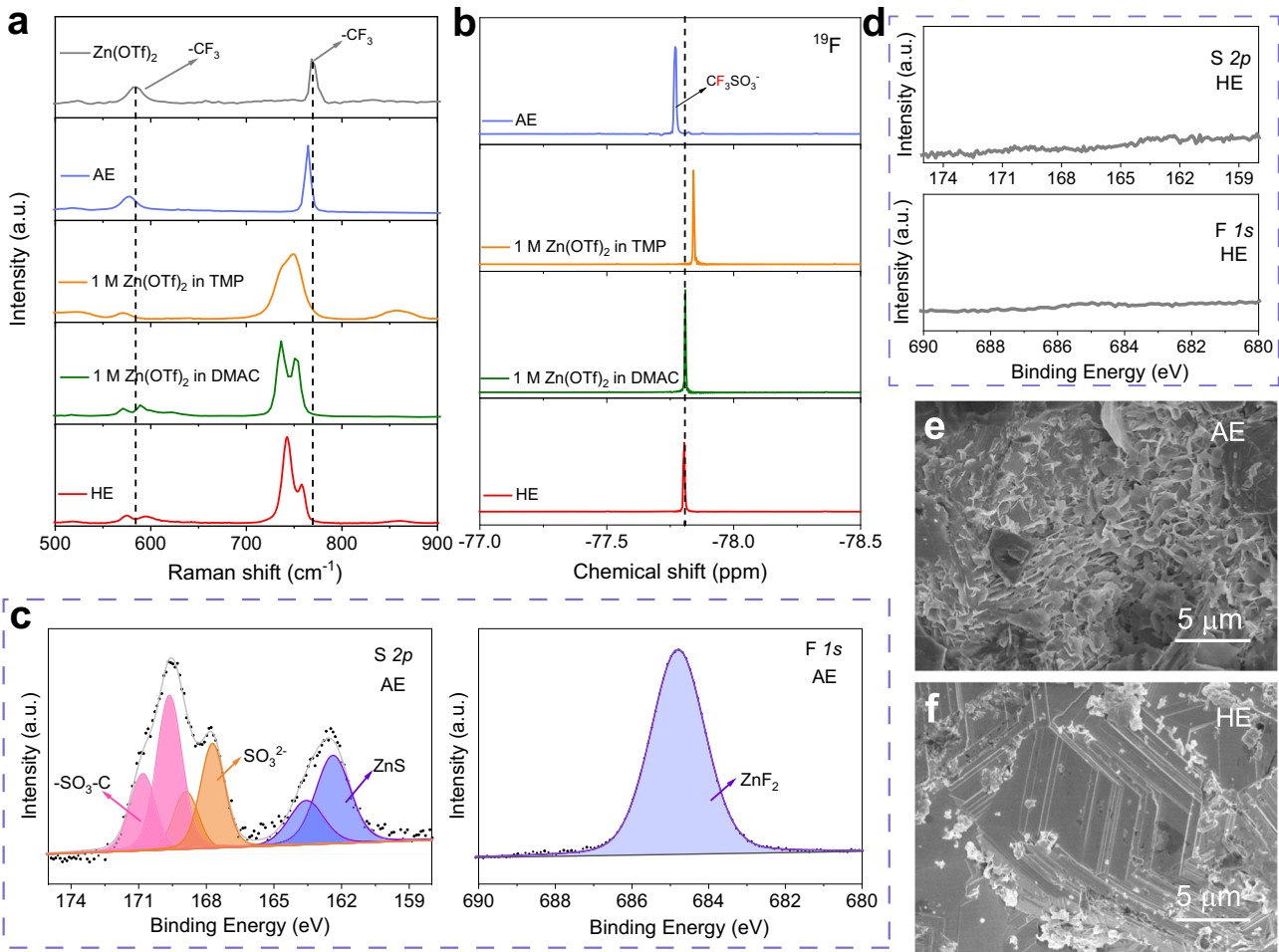

**Fig. 3 | Analyses of side reactions between electrolyte and Zn anode. a** Raman spectra for Zn(OTf)$_2$ and selected electrolytes. **b** $^{19}$F NMR spectra for selected electrolytes; High-resolution XPS spectra for Zn foils following working in (**c**) AE and (**d**) HE for 100 cycles. **e**, **f** Morphology for Zn foil following working in AE/HE for 100 cycles.

the Zn foil cycled in AE has a 'pulverized' surface which may result in the loss of active metallic Zn, Fig. 3e.

NVO is a promising cathode for ZIB because the multivalence of vanadium contributes to high capacity. More importantly, NVO has a large interlayer distance, 0.77 nm, allowing for Zn$^{2+}$ insertion/extraction. In addition, NVO can be synthesized on a large scale via a low-cost liquid-solid stirring strategy under ambient conditions, providing potential for real applications. Therefore, NVO is selected as the cathode to determine whether HE is compatible in the full cell. The synthesized NVO has a nano-belt morphology (Supplementary Fig. S19) and its interplanar spacing of the (001) plane is measured to be ca. 0.77 nm by the high-resolution transmission electron microscopy (TEM) (Supplementary Fig. S20b). The elemental mapping images (Supplementary Fig. S20d) confirm the uniform distribution of Na, V, and O. The 3-electrode CV curve reveals that there are two oxidation peaks and three reduction peaks during the charging/discharging, and all redox reactions occur between 0.3 V and 1.6 V (Supplementary Fig. S21). Besides, the electrochemical reactions of NVO electrode are composited with ionic diffusion control and pseudo-capacitance control (Supplementary Fig. S22). The V 2p XPS spectra confirm the valence variation for vanadium during discharge/charge. In pristine NVO electrode the majority of vanadium is at 5+ valence and a small portion at 4+ valence. When discharged at 0.3 V the intensity of the V$^{5+}$ signal declines while that for V$^{4+}$ increases. The peak intensity reversed following charging back to 1.6 V, confirming the electrochemical reversibility of NVO (Supplementary Fig. S23).

At low specific current, Zn‖NVO cells with AE and HE exhibit similar energy density, however, the battery with AE exhibits superiority at high current densities (Supplementary Fig. S24). To evaluate the energy density at the electrode scale, Zn‖NVO cells are assembled with a thick cathode and a thin Zn foil, in which the cathode mass loading is 9.3 mg cm$^{-2}$ and the areal capacity ratio of negative to positive electrodes (N/P ratio) is 5. Because of high ionic conductivity of AE, the Zn‖NVO cell exhibits better rate performance, with ca. 80 mAh g$^{-1}$ at 5 A g$^{-1}$, as is seen in Fig. 4a. The Zn‖NVO cell using HE does not exhibit an advantage in specific capacity, especially when the applied specific current increases. The energy density and power density of Zn‖NVO batteries are given according to the rate performance, calculated with the mass of the cathode and anode (Fig. 4b). The battery in AE exhibits a superior energy density of 86.1 Wh kg$^{-1}$ and greater power energy density of 1019 W kg$^{-1}$, while in HE, the energy density is 77 Wh kg$^{-1}$ and the power maximum is 665 W kg$^{-1}$. The energy density of Zn‖NVO battery in HE surpasses that of the majority of aqueous Li-ion and Na-ion batteries, and approaches the value for the state-of-the-art Li-ion batteries (Supplementary Table S2 and Fig. S25). However, Zn‖NVO battery is apparently more competitive than aqueous Li-ion batteries when affordability is taken into account. Although HE has no superiority in rate performance, it significantly improves stability of Zn‖NVO cells. At a low specific current of 0.1 A g$^{-1}$, the specific capacity for the aqueous Zn‖NVO cell faded significantly within 200 cycles, while in HE the Zn‖NVO cell maintained a specific capacity of 164 mAh g$^{-1}$ following 700 cycles

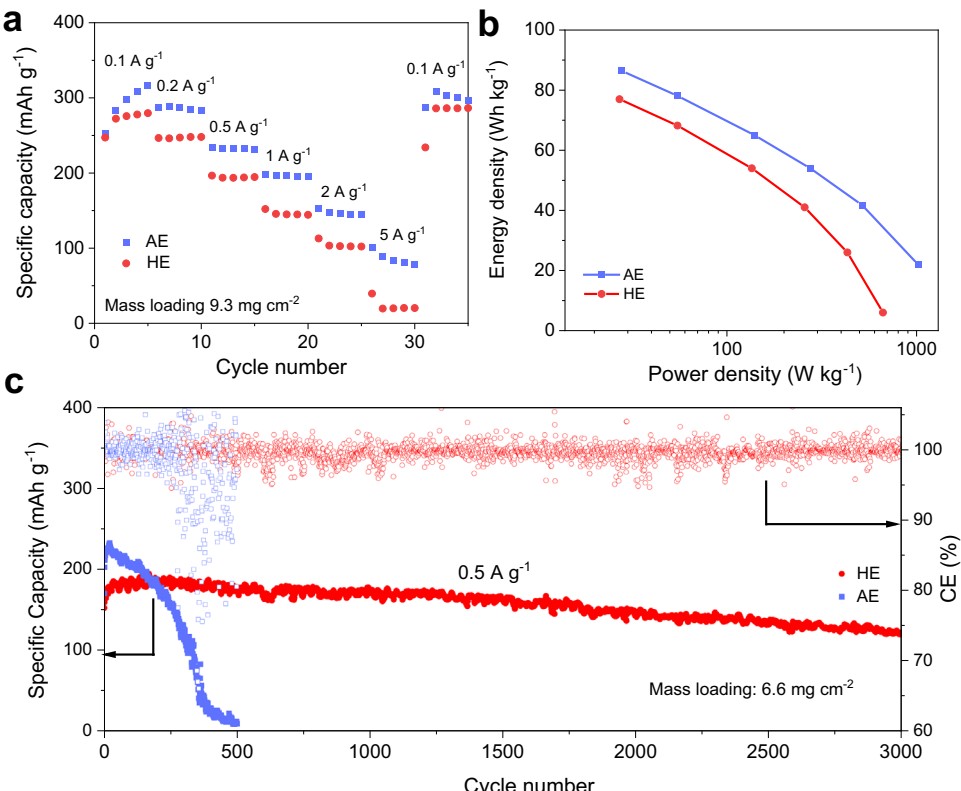

**Fig. 4 | Electrochemical performance of Zn||NVO cells. a** Rate performance. **b** Energy and power density for Zn||NVO cells calculated based on the mass of cathode and anode active materials. **c** Cyclic stability and CE of Zn||NVO cells at a specific current of 0.5 A g⁻¹.

(Supplementary Fig. S26). At a specific current of 0.5 A g⁻¹, in AE, the NVO cathode exhibits a rapid increase in specific capacity in the first 20 cycles from 170 to 231 mAh g⁻¹, and an apparent decline in subsequent cycling, to fail completely in the following 400 cycles, Fig. 4c. The corresponding charge-discharge curves exhibit that the voltage decays significantly following 300 cycles (Supplementary Fig. S27). In contrast, the specific capacity for the NVO cathode in HE remains stable and has a lifespan of 3000 cycles. To determine the reason for aqueous Zn||NVO battery failure, NVO electrodes were extracted when capacity decreased significantly and reassembled with fresh Zn anode and sufficient electrolyte. It was found that following the refresh, specific capacity for NVO electrodes could not be recovered to that of its initial state, evidencing that cathode degradation significantly affects battery performance in AE (Supplementary Fig. S28).

We hypothesized that the significantly improved reversibility of the NVO cathode in HE is related to structural stability. To confirm this, *in operando* synchrotron-based X-ray powder diffraction (XRPD) was used to monitor lattice changes in NVO during discharge and charge, respectively, AE and HE. Figure 5a, b presents the (204) reflection as a contour-plot with color-intensity and corresponding electrochemical curves. The (204) plane was selected for analyses because (204) reflection is the strongest and clearest peak obtained *in operando* conditions (Supplementary Fig. S31). For the NVO cathode working in AE, its (204) reflection shifts to lower angles when the Zn||NVO battery discharges from open-circuit voltage to 0.3 V. In this, Zn²⁺ ions intercalate into the layer of NVO accompanied by the continuous increase of lattice parameter and cell volume. A reversed behavior is observed during charging. The peak moves right-wards during charging, corresponding to Zn²⁺ ions extraction and a decrease in the d-spacing. The peak shift is continuous and reversible, evidencing solid-solution-like behavior during Zn²⁺ insertion/extraction. Notably, at deep discharge state, the (204) reflection intensity is greatly reduced, implying the

order-disorder transition at high Zn concentrations. A similar trend was found in the NVO cathode using HE as the electrolyte. In HE, the lattice change was relatively minor and the peak intensity remained nearly unchanged across the charge-discharge processes, confirming that the structural changes in NVO are significantly less than with AE and absent of order-disorder transition. These differences likely originate from the solvation structure of electrolytes. Water molecules are co-intercalated with the Zn²⁺ ion into the structure of NVO during discharge, which circumvents the desolvation energy penalty during (de)intercalation and shields the electrostatic interaction between the Zn²⁺ ion and the host matrix[33,34]. Water molecules serve as 'pillars' to increase the layer spacing and therefore accelerate reaction kinetics and improve Zn²⁺ store ability, which accounts for the higher initial specific capacity and the rapid activation of the NVO cathode in AE. However, the large radius of the hydrated Zn²⁺ clusters (Zn(H₂O)ₙ²⁺), ca. 5.5 Å[9], results in significant lattice distortion of the host material and leads to structural collapse when the lattice stress exceeds its limit[35]. This vulnerable structure of NVO results in rapid decline in specific capacity and a limited lifespan. The outcome is different when the NVO cathode works in HE, because, unlike AE, where the first hydration shell is composed of six water molecules, the number of water in the solvation cluster of HE is reduced to 3.7. DMAC and TMP in the solvation cluster exert strong attraction to H₂O, meaning fewer water molecules combine with Zn²⁺ when hydrated Zn²⁺ enters the structure of NVO. As a result, the smaller size of guest species in NVO crystal brings less lattice distortion and ensures better structural stability during long-term cycling. Figure 5d presents the XRPD pattern for NVO electrode following cycling in HE for 50, 100, and 200 cycles, evidencing that the structure of NVO is maintained during cycling and no side product(s) is found. However, in AE, Zn₄SO₄(OH)₆·4H₂O formed and the (001) reflection weakened and almost disappeared following 200 cycles, Fig. 5c. CV curves of Zn||NVO cell (Fig. 6a) show

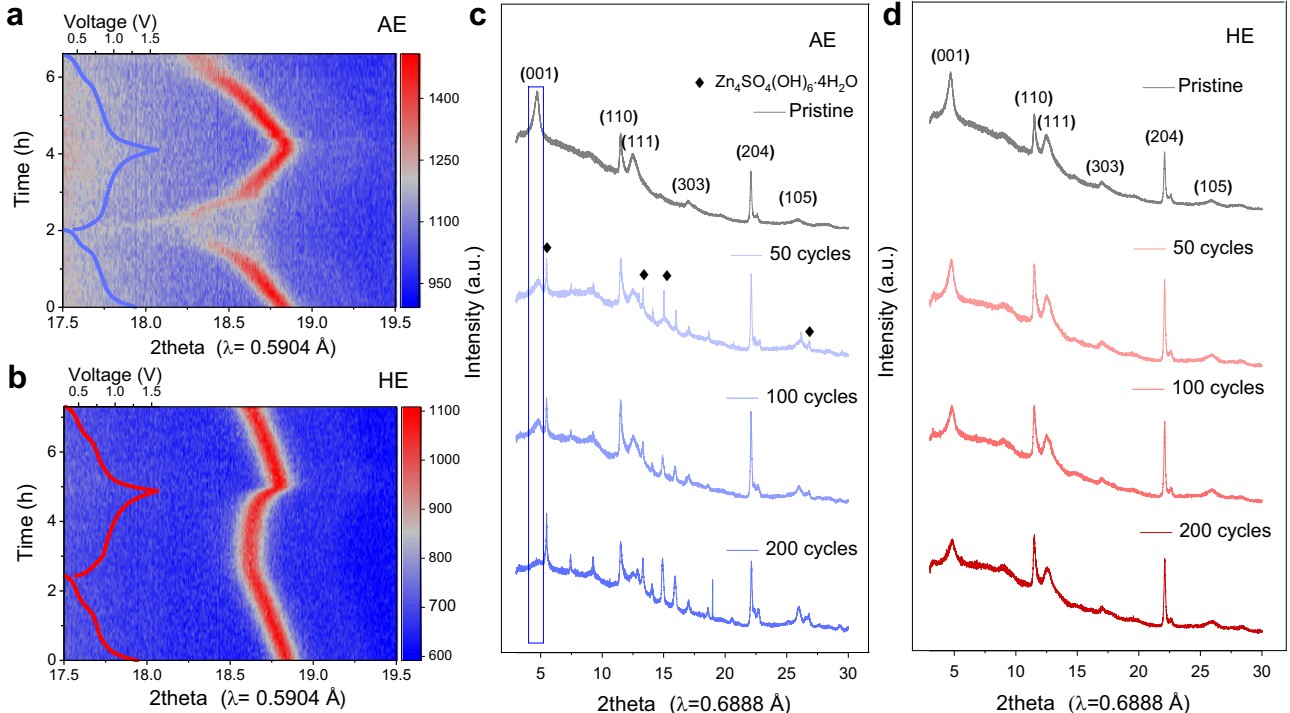

**Fig. 5 | Structure evolution of NVO electrode.** Contour plot for NVO (204) reflection, using **a** AE and **b** HE as electrolyte, in which the (204) reflection evolutes along with corresponding charge/discharge curve; synchrotron-based XRPD patterns for NVO electrodes following cycling in **c** AE and **d** HE.

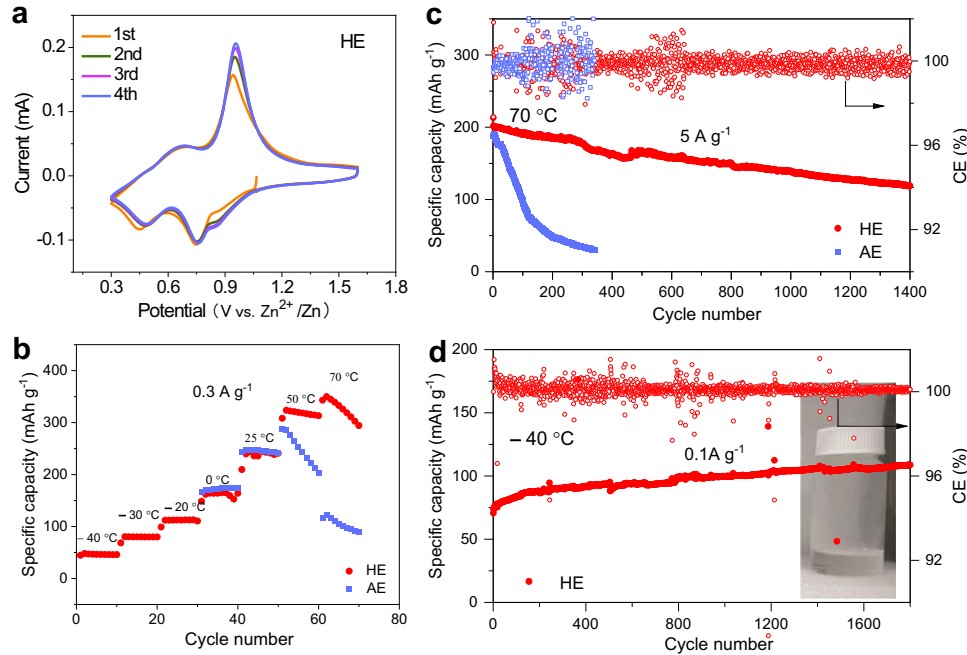

**Fig. 6 | CV curves of NVO electrode and temperature adaptability of Zn‖NVO cells in AE and HE. a** 3-electrode CV curves for NVO electrode tested in HE. **b** The specific capacity of NVO electrode tested at temperatures ranging from −40 °C to 70 °C. **c** Cyclic performance for Zn‖NVO cells at 70 °C with AE and HE as electrolyte. **d** Cyclic performance at −40 °C in the presence of HE (HE is liquid at −40 °C).

that the redox reactions are completely reversible in HE, better than that in AE (Supplementary Fig. S32). DMAC/TMP/H$_2$O hybrid solvent in a volume ratio of 5:2:3 is compatible with salts including zinc tri-fluoromethanesulfonate (Zn(TFSI)$_2$) and Zn(ClO$_4$)$_2$. In hybrid electrolytes using Zn(TFSI)$_2$ or ZnClO$_4$ as salt, the NVO cathode maintains specific capacity without decay for >1200 cylces, and the CE for Zn

plating/stripping is boosted compared with that in corresponding aqueous electrolyte (Supplementary Fig. S33).

To determine the compatibility of HE with high-voltage cathode, zinc hexacyanoferrates (Zn$_3$(Fe(CN)$_6$)$_2$), was selected that exhibits an operation voltage of 1.8 to 1.6 V when combined with the Zn anode. The initial specific capacity for Zn‖Zn$_3$(Fe(CN)$_6$)$_2$ cells is ca. 60 mAh g$^{-1}$

in both electrolytes (Supplementary Fig. S35a). However, it decreases rapidly in AE accompanied by voltage decay (Supplementary Figs. S35b, c). The output voltage for $Zn_3(Fe(CN)_6)_2$ reduces to 1.6 ~ 1.4 V following 150 cycles in AE (Supplementary Fig. S35d). This is attributed to material degradation and less stability of $H_2O$ at high voltage during charging, that causes relatively low CE. In contrast, the $Zn||Zn_3(Fe(CN)_6)_2$ cell exhibits better stability in HE and the high-voltage character is well maintained following 150 cycles, evidencing that HE is compatible with the high-voltage cathode.

To investigate the operating temperature of AE and HE, differential scanning calorimetry (DSC) and thermogravimetric analysis (TGA) were conducted. For AE, there is an exothermic peak at ca. −29 °C corresponding to the crystallization of electrolyte and an endothermic peak at −4 °C corresponding to the melting point. In contrast, no exothermic/endothermic peaks are observed for HE even with the temperature cooling down to −80 °C (Supplementary Fig. S37). The TGA curves show that AE loses mass faster than HE because of fast $H_2O$ volatilization. While in HE, the strong interactions between $H_2O$ and DMAC/TMP effectively reduce volatilization effects, increasing the stability of HE at high temperatures. The ionic conductivity and viscosity of AE and HE are given in Supplementary Table S3. AE has advantages in both conductivity and viscosity at temperatures >0 °C. A Walden plot can be drawn via calculating molar conductivity and viscosity, in which it is evidenced that $Zn(OTf)_2$ salt is well-dissociated in these two electrolytes (Supplementary Fig. S38).

To assess the performance of the electrolytes in a high- or low-temperature environment, Zn||NVO batteries using AE and HE as the electrolyte were tested at various temperatures (Fig. 6b). The Zn||NVO battery in the presence of AE exhibited a rapid capacity degradation, Fig. 6c, and significantly increased polarization (Supplementary Fig. S39a) at 70 °C. In contrast, the Zn||NVO battery with HE as electrolyte exhibited an initial specific capacity of 207 mAh g⁻¹ and maintained a reversible capacity of 120 mAh g⁻¹ following 1400 cycles. The significantly boosted thermostability of HE is attributed to participation of DMAC and TMP in the electrolyte, that interacts strongly with $H_2O$, strengthening the O−H bonds of water and therefore reducing the decomposition of $H_2O$. In addition, HE exhibited a freeze-tolerance at sub-zero temperatures. At −20 °C, the Zn||NVO battery exhibited a capacity of ca. 140 mAh g⁻¹ without apparent capacity fading for 2000 cycles (Supplementary Fig. S40), confirming that HE contributes to a highly reversible battery. At as low as −40 °C, the HE remains in liquid state and the Zn||NVO battery exhibted a capacity of ca. 85 mAh g⁻¹, Fig. 6d. AE failed to support the operation of the Zn||NVO battery because it freezes at sub-zero temperatures (Supplementary Fig. S41). The mechanism for anti-freezing is attributed to the strong dipole-dipole interaction between DMAC/TMP and water molecules that breaks the extension of the hydrogen-bonded network of water, reducing the freezing point of the electrolyte.

A comparison of state-of-the-art electrolytes is listed in Supplementary Table S4. For non-aqueous electrolytes, the choices of compatible cathode materials are quite limited and the electrochemical properties, such as specific capacity and lifespan, of these cathodes are inferior compared with aqueous batteries[14,19]. As for those aqueous-based electrolytes, in which water accounts for >50% by volume, the low-rate performance and high-temperature performance of batteries become the shortcomings[23,36,37], because the interactions between water and non-aqueous solvents are not sufficient to suppress the water activity. In our design, we emphasized the impact of water content on the specific capacity and stability of the NVO cathode and further demonstrated that the stability is highly related to the structural evolution of the cathode during battery operation. On comparison, the Zn||NVO battery in HE has advantages at a low cycling specific current (0.1 A g⁻¹), and superior temperature adaptability.

## Discussion

Polar solvents dimethylacetamide (DMAC) and trimethyl phosphate (TMP) in a hybrid electrolyte (HE) exert strong dipole-dipole interaction with $H_2O$ to increase the electron density of water protons making the $H_2O$ dissociation less thermodynamically favorable and therefore strengthening O−H bonds of water. $Zn^{2+}$ plating in HE preferentially orientates the (002) plane because of the lower surface energy of this plane compared with (100) and (101) planes. With confined water activity and favorable orientation, a Zn anode in HE exhibited a high reversibility and ultra-long life in which CE was 99.5% for 2000 cycles, and Zn||Zn symmetric cells ran for 1600 h at a current density 5 mA cm⁻² and areal capacity 5 mAh cm⁻². The unique solvation structure and strong attraction between DMAC/TMP and $H_2O$ leads to a reduced size of hydrated $Zn^{2+}$ that maintained the structural stability of NVO to exhibit 3000 cycles for the Zn||NVO cell. The strong interaction between solvents expanded temperature adaptability of electrolyte with the result that the Zn||NVO cell ran for 1400 cycles at 70 °C, and for >1800 cycles at −40 °C. Findings will be of immediate benefit in practical design for highly reversible aqueous zinc ion batteries (ZIBs) and therefore of wide interest to researchers and manufacturers.

## Methods
### Chemicals
Dimethylacetamide (DMAC, 99.8%), trimethyl phosphate (TMP, 99%), zinc trifluoromethanesulfonate ($Zn(OTf)_2$, 98%), vanadium pentoxide ($V_2O_5$, 98%), potassium hexacyanoferrate(III) ($K_3Fe(CN)_6$, 99.98%), zinc sulfate ($ZnSO_4·7H_2O$, 99%), zinc perchlorate hexahydrate ($Zn(ClO_4)_2·6H_2O$, 99%), and zinc trifluoromethanesulfonate ($Zn(TSFI)_2$, 98%) were purchased from Sigma-Aldrich.

### NVO synthesis
To 100 mL, 2 M L⁻¹ NaCl solution add 3 g commercial $V_2O_5$ powder followed by magnetic stirring at 400 rpm for 72 h. As obtained orange-color powders are centrifuged and washed with deionized water and ethanol five times and then dried at 80 °C for 10 h.

### $Zn_3(Fe(CN)_6)_2$ synthesis
50 mL of 0.1 M $ZnSO_4·7H_2O$ and 50 mL of 0.05 M $K_3Fe(CN)_6$ were mixed with 25 ml deionized water, followed by heating at 60 °C under vigorous stirring for 5 h to complete the reaction. The precipitate was rinsed and centrifuged five times to remove the residues. Then, the product was finally dried at 70 °C for 12 h.

### Characterization
XRD measurements on NVO powders were determined with a Rigaku MiniFlex600 with Cu Kα radiation. Morphology of Zn foils and NVO powders were evidenced with a JEOL JSM-7500FA field emission scanning electron microscope. A Thermo Scientific Nexsa X-Ray Photoelectron Spectrometer System was used to determine XPS spectra of cycled Zn foils. FT-IR spectra and Raman spectra, were determined, respectively, with a PerkinElmer Frontiers instrument with an attenuated total reflectance (ATR) attachment, and a Raman spectrometer (Horiba LabRam Evolution). In situ optical observation of the Zn deposition process in various electrolytes was conducted with a customized cell (EL-CELL). LSV was conducted on a VMP3 instrument, with 10 μm Pt ultramicroelectrode as the working electrode, Pt plate electrode as the counter electrode and Ag/AgCl electrode as the reference electrode. CV was conducted with a 3-electrode EL-CELL using well-polished Zn as the reference electrode. NMR testing was carried out with a Bruker Avance Neo 500-MHz NMR spectrometer using a cryoprobe. *In operando* mechanistic studies on the NVO electrode were conducted with an *in operando* synchrotron-based X-ray diffraction at the Powder Diffraction beamline at the Australian Synchrotron. CR2032 coin cells with 4mm-diameter windows were

especially prepared to ensure synchrotron beam transmission, and; Kapton tape was used to seal the holes to avoid electrolyte leaking. The wavelength of the synchrotron X-ray beam was 0.59040(1) Å or 0.68880(1) Å, as determined using $La^{11}B_6$ NIST standard reference material 660b. The customized coin cells were tested at a specific current of 100 mA g$^{-1}$ between 0.3 and 1.6 V (vs. Zn/Zn$^{2+}$). The diffraction patterns were recorded with an exposure time of 3 min by an MYTHEN microstrip detector. TEM images were collected on FEI Titan Themis 80-200. The ionic conductivity of electrolytes was measured with a Thermo Scientific Orion electrochemistry meter, and the viscosity is measured with capillary viscometers (Huanguang Brand, Zhejiang, China) under certain temperatures. The Mettler Toledo thermogravimetric analyser/differential scanning calorimeter 3+ is used for TGA and DSC measurement.

### Electrochemical testing

Electrochemical data were collected from the Neware battery testers. To prepare the NVO and $Zn_3(Fe(CN)_6)_2$ electrodes, a slurry composed of active materials, Super P and polyvinylidene difluoride (PVDF) at a mass ratio of 7:1.5:1.5 was cast onto Ti-foil and followed with drying at 80 °C for 12 h. Then, the electrode was cut into small disks with a diameter of 0.95 cm. The mass loading of the NVO cathode ranges from 0.6–9.3 mg cm$^{-2}$ and that of $Zn_3(Fe(CN)_6)_2$ electrode is 1.2 mg cm$^{-2}$. CR2032-type coin cells were used in all electrochemical tests. Clean Zn foil (99.99%, 100 μm) is used as the anode without further treatment. For batteries using aqueous electrolyte (AE), a glass-fiber membrane (740 μm) was used as the separator, and for those with hybrid electrolyte (HE) or non-aqueous electrolytes, nylon 66 membrane (130 μm) is used. The glass-fiber membrane was purchased from Filtech Pty Ltd, while the nylon 66 membrane was purchased from Shanghai Xinya Purification Equipment Co., Ltd. The batteries are tested at room temperature unless otherwise specified. High- and low-temperature performance of batteries are tested in a temperature-controlled chamber. The cut-off potentials for Zn||NVO cells are set to be 0.3 V and 1.6 V, and that for Zn||$Zn_3(Fe(CN)_6)_2$ cells are 1 V and 2 V. To evaluate the energy and powder density of Zn||NVO cell, thick NVO electrode with active material loading of ca. 9.3 mg cm$^{-2}$ and thin Zn foil (10 μm) are used for battery assembly. The energy and powder density values are calculated based on the mass of cathode and anode materials.

### MD simulation

All of the ion parameters were obtained from the literature[38], namely GAFF2 force field. Other molecules were optimized via gaussian 16 package at a level of B3LYP/def2tzvp firstly and vibration analyses were performed at the same level to ensure that there were no virtual frequencies. The ACPYPE webserver was used to obtain the GAFF2 force-field topology file and Packmol software[39] used for construction of the model. Simulation was as follows: the 5000-step steepest descent and 5000-step conjugate gradient method were used to obviate unreasonable contact with the system; NPT ensemble was used to pre-equilibrate the system, and V-rescale temperature coupling and Parrinello-Rahman pressure coupling were used to control the temperature to 298 K; pressure was maintained at 1 atm, non-bonding cutoff radius was 1.2 nm; integration step was 2 fs. 30 ns simulations were performed in which bond length and angle were constrained by the LINCS algorithm. The two-way intercept was set to 1.2 nm, van der Waals interaction and the long-distance electrostatic interaction was set via the particle-mesh Ewald method. The trajectory file during simulation was saved each 10.0 ps.

### DFT computation

All DFT computations were performed via Vienna ab initio simulation package (VASP) developed by Fakultät für Physik[40,41]. A plane wave energy cut-off of 500 eV was used. The generalized gradient approximation (GGA) and the Perdew−Burke−Ernzerhof (PBE) XC functional combined with the projector augmented wave (PAW) method were used to describe the exchange-correlation functional[41,42]. The MonkhorstPack scheme was used to generate the k-point sampling grids within the Brillouin zone[40]. Energy differences of $1 \times 10^{-5}$ eV/atom and energy difference gradients of −0.01 eV/Å were used as convergence criteria[43].

For deprotonation reaction computation, we picked the Zn$^{2+}$ cluster structures from the MD simulation for further analysis using DFT. The dispersion correction was used by employing the DFT-D3 method[44,45]. The deprotonation energy for $H_2O$ was computed from the following:

$$E_{\text{deprotonation}} = E_{\text{(cluster)}} - \frac{1}{2A}(E_{H_2}) - E_{\text{(deprotonated cluster)}} \quad (1)$$

where $E_{H_2}$ is the energy of $H_2$ used for the standard hydrogen electrode scale ($H^+ + e^- \rightleftharpoons \frac{1}{2} H_2$), $E_{\text{(cluster)}}$ and $E_{\text{(deprotonated cluster)}}$ are the energies of the cluster in the presence and absence of $H^+$, respectively.

For Zn bulk computations, a $9 \times 9 \times 9$ grid was used for the supercells of Zn (100), Zn (101), Zn (002), and for the solvation model covered structures, the grid was set as $3 \times 3 \times 1$. The solvation model were obtained using the MS FORCITE module, and FORCITE optimizations are performed first, which was a molecular mechanics module for potential energy and geometry optimization calculations of arbitrary molecular and periodic systems using classical mechanics[45]. The surface energy for Zn (100), Zn (101), Zn (002) surfaces ($\gamma_s$) was defined by:

$$\gamma_s = \frac{1}{2A}(E_s^{\text{unrelax}} - NE_b) + \frac{1}{A}(E_s^{\text{relax}} - E_s^{\text{unrelax}}) \quad (2)$$

where $A$ is the area of the surface considered, $E_s^{\text{relax}}$ and $E_s^{\text{unrelax}}$ energies for, respectively, relaxed and unrelaxed surfaces, N number of atoms in the slab and E$_b$ bulk energy per atom[46,47]. For the solvation model the surface energy for Zn (100), Zn (101), Zn (002) surfaces ($\gamma_{\text{sol}}$) was defined as:

$$\gamma_{\text{sol}} = \gamma_s + \frac{E_{\text{sol}}}{A} \quad (3)$$

where $E_{\text{sol}}$ is given by:

$$E_{\text{sol}} = (E_{\text{slab−sol}} - E_{\text{slab}} - E_{\text{sol}}) \quad (4)$$

where $E_{\text{slab-sol}}$ is energy of the surface covered with the solvation molecules, $E_{\text{slab}}$ energy for the Zn (100), Zn (101), or Zn (002) clean surface, and $E_{\text{sol}}$ energy for the solvated models.

## Data availability

The data generated in this study are provided in the Supplementary Information/Source data file. Source data are provided with this paper.

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

## Acknowledgements

The authors gratefully acknowledge the financial support provided by the Australian Research Council (DP210101486 Z.G., DP200101862 Z.G., and FL210100050 Z.G.). Y.W. acknowledges the Chinese Scholarship Council for scholarship support (No. 201808440447 Y.W.). The authors acknowledge the operational support from ANSTO staff for synchrotron-based characterizations (Awarded beamtime: M17943 W.K., M18569 G.M., and M18654. S.L.). J.A.Y. acknowledges the assistance of resources and services from the National Computational Infrastructure (NCI), which is supported by the Australian Government.

## Author contributions

Conceptualization: Z.G., B.L., Y.W., and Z.W. Experimental design and investigation: Y.W. and Z.W. Data analyses: Y.W., Z.W., G.L., S.L., and Y.F. Characterization of synchrotron-based XRPD: Z.W., A.A., and W.K.P. Theoretical simulation: J.A.Y. and J.D. NMR characterization: W.L. Writing—original draft: Y.W. and Z.W. Writing—review & editing: Y.W., Z.W., Z.G., K.D., and B.L.

## Competing interests

The authors declare no competing interests.
