## [Peer Review File · Nature Communications]

Solvent control of water O–H bonds for highly reversible zinc ion batteriesREVIEWER COMMENTS

Reviewer #1 (Remarks to the Author):

This work reports a ternary solvent component for ZIBs to tackle with the current challenges. The proposed hybrid electrolyte (HE) allows reliable cycle performance compared with aqueous electrolyte (AE). More importantly, the underlying mechanism is discussed and supported by various characterization tools. Overall, the reviewer recommends this work to be published in Nature Communications. Some suggestions are provided for the authors to improve the MS:

1. There are many sentences should be supported by proper references in the Introduction part.
2. Several related works (not the reviewer's work) are suggested to be discussed in the MS, such as DMAC used in Zn-based batteries (ACS Appl. Mater. Interfaces 2021, 13, 39, 46634–46643), triethyl phosphate (TEP) used in Zn-based batteries (Angew. Chem. Int. Ed. 2019, 58, 2760–2764), and TMP used in LIBs (J. Electrochem. Soc. 148 A1058; ACS Appl. Mater. Interfaces 2019, 11, 39, 35770–35776; Chem. Commun., 2018, 54, 4453–4456). Also, could the authors highlight the novelty of the current work by comparing with the above papers?
3. Is 5:2:3 the optimized ratio for DMAC/TMP/H₂O solvent? For Zn salts, is the proposed ternary solvent compatible with ZnSO₄ and Zn(TFSI)₂? The former is the most commonly used Zn salt (and cheap), while the later has been found to enable promising Zn²⁺ storage properties for V-based and Mn-based cathodes.
4. (002) preferred orientation is proposed to be a key factor in deposited Zn. However, XRD results were not provided.
5. Are there any Na residuals in the V₂O₅ sample? Mixing NaCl and V₂O₅ usually results in the precipitation of NaV₃O₈ (for instance, Nat. Commun., 2018, 9:1656; ACS Appl. Energy Mater. 2018, 1, 6401), but V₂O₅ was obtained in the current work. Could the authors comment on this?
6. Pristine V₂O₅ has predominant (001) diffraction peak without noticeable (020) reflection. However, in situ XRD data only show the changes of (020) reflection.
7. The high reversibility of V₂O₅ in HE is attributed to the good structural stability (compared with AE). Could the authors provide XRD data before and after cycling in AE and HE to support this statement.
8. One of the merits of HE is wider ESW. But the redox reactions of V-based materials are relatively low. Could the authors demonstrate the performance of other cathode materials with high operating voltage?
9. What is the ionic conductivity of the HE (compared with AE)? Rate performance is also important for batteries. Moreover, it is recommended to provide a comparison table including the electrochemical performance of the current work and reported papers.
10. The experimental details should be provided, such as grade of Zn anode (thickness, purity...), supplier of battery separators, mass loading for V₂O₅ cathode.

Reviewer #2 (Remarks to the Author):

Through this good scientific paper, the authors present a battery based on a new aqueous electrolyte, mainly by playing on the control of the O-H bond. Despite the interest I had in reading the scientific paper, some weaknesses are present here and do not allow a firm conclusion. 2/5 of the identified weaknesses are significant to accept the present work (comments (iii) and (iv)).

(i) Lack of standard measurement of formulated electrolytes. It is now recognized that impurities in formulated electrolytes in batteries influence performance and, at the same time, confound measurement. Thus the characterization of electrolytes (as well as electrodes) must be characterized with rigor and detail. In the present case is missing, to my knowledge:

- (i).1 Thermal measurements: DSC, TGA
- (i). 2 Ionic conductivity and viscosity (Walden Plot).

(ii) The cell voltage measurement is strongly related to the ESW. In this case, the effect on the positive electrode is unambiguously observed. However, no measurements at the negative electrode are made to identify the ESW. This is a significant shortcoming.

(ii).1 There is a need to study ESW at both the positive and negative electrodes. The authors should ideally carry out this study in LSV via ultramicroelectrodes to avoid the ohmic drop. But already providing a measurement of the limit in negative potential is necessary.

(ii). 2 In figure S22, the window of the GCD measurement is not convincing for a two-electrode system; the authors vary the voltage from 0.3 to 1.5. The value of 0.3 deserves justification.

(iii) This paper does not present performances measurements. Today, it seems necessary that device metrics be constructed with reproducibility and comparison (see doi.org/10.1002/batt.202100154 AND <https://doi.org/10.1038/s41467-022-29257-w>). In this respect, plotting a Ragon Plot (normalized by mass and then by volume) seems essential to conclude the interest of the electrolyte proposed here. To my knowledge, the present article does not provide energy density and maximum power values. This is one of the major criticisms of the present work: NO performances measurement allows us to conclude on the real interest of the work:

(iii). 1 Lack of Ragon Plot

(iii). 2 Absence of a measure of Energy Density.

(iii). 3 Lack of a Power Max.

(iii). 4 Absence of different C-rate measurements.

(iii). 5 In this context, the measurement in Fig. 4 opens a substantial ambiguity and weakens the seriousness of this paper. Indeed, Fig. 4d and Fig. 4e are not made at the same C-rate (current range). This is understandable because the temperature is not the same, but if the device was operational at a wide range of temperatures, it should cycle at the same C-rate. Otherwise, it would bias the result to show two different C-rates.

(iv) Information related to the V2O5 cathode. Missing information limits the interest of this electrode.

(iv). 1 The current collector for the V2O5 electrode is not identified. However, a study of its corrosion (even qualitative) is necessary for the seriousness of the study.

(iv).2 The justification of the choice of V2O5 does not open the interest of the study. The authors should justify the choice of this electrode material.

(iv).3 A 3-electrode study of V2O5, depending on the nature of the electrolyte, would be a necessary added value to understanding the electrochemical dynamics at the cathode. Also, for the Zn anode.

(v) Minor remarks:

(v). 1 The authors write OTF-1 instead of OTF1-, in the same way that the Zinc cation is written Zn²⁺ and not Zn+2; the authors should correct this minor error.

(v).2 Figures 2a, b, c, and d, and figures S12 and S13 should be made more prominent as it is difficult to see the SEM images.

Based on the five family comments, it is difficult to conclude the genuine interest in the present article: performance measures are missing, and information limits the reader's curiosity.

Reviewer #3 (Remarks to the Author):

In this research, the authors tried to demonstrate the enhanced electrochemical properties of Zn-V2O5 battery through controlling the electrolyte composition. However, it seems some critical issues weren't solved to have strong recommendation for publication at Nature Communications.

1. There is no detailed reason why the authors focused on hybrid electrolyte of dimethylacetamide and trimethyl phosphate. Actually, it was reported usage of dimethylacetamide and trimethyl phosphate can lead to outstanding electrochemical performances of Zn-ion batteries, respectively. (ACS Appl. Mater. Interfaces 2021, 13, 39, 46634–46643, Angew. Chem. Int. Ed., 2019, 58, 2760–2764) Moreover, it seems the related references are not sufficiently cited in this manuscript.

2. It is also required to perform the electrochemical tests under other DMAC/TMP/H₂O electrolytes with different volume ratio. Although each case of TMP/H₂O and DMAC/H₂O were studied on an effect of different volume ratio, it is insufficient to explain the reason on usage of DMAC/TMP/H₂O (5:2:3 by volume). The authors should show why the volume ratio of 5:2:3 (=DMAC/TMP/H₂O)

was used as the optimal electrolyte composition for this research.

3. Actually, the researches on Zn-V₂O₅ battery was already reported at many papers. Thus, it is also very important to confirm the target material can exhibit the general electrochemical performances of Zn-V₂O₅ battery. However, the author didn't present the charge/discharge curves and CV curves of Zn-V₂O₅ battery measured at a low current density with room temperature. Moreover, there is no data showing the redox reaction of vanadium ions during charge and discharge.

4. Several SEM results show the electrolyte affected the morphology of Zn-foil after cycling. These results imply the main effect of the electrolyte for electrochemical performances of Zn-V₂O₅ battery may be attributed to the morphology of Zn-foil, not structural evolution of V₂O₅. The authors should explain what is the main effect to determine the enhanced electrochemical performances of Zn-V₂O₅ battery between the morphology of Zn-foil and structural evolution of V₂O₅ in detail.

RESPONSE TO REVIEWS

Response to Reviewer #1

Reviewer's Remarks to Authors

This work reports a ternary solvent component for ZIBs to tackle with the current challenges. The proposed hybrid electrolyte (HE) allows reliable cycle performance compared with aqueous electrolyte (AE). More importantly, the underlying mechanism is discussed and supported by various characterization tools. Overall, the reviewer recommends this work to be published in Nature Communications. Some suggestions are provided for the authors to improve the MS:

Response

We thank Reviewer #1 for his/her valuable comments and recommendation for publication.

Comment 1-1

There are many sentences should be supported by proper references in the Introduction part.

Response

Thanks for your suggestion. We have added 19 references in the Introduction.

Comment 1-2

*Several related works (not the reviewer's work) are suggested to be discussed in the MS, such as DMAC used in Zn-based batteries (*ACS Appl. Mater. Interfaces* 2021, 13, 39, 46634–46643), triethyl phosphate (TEP) used in Zn-based batteries (*Angew. Chem. Int. Ed.* 2019, 58, 2760–2764), and TMP used in LIBs (*J. Electrochem. Soc.* 148 A1058; *ACS Appl. Mater. Interfaces* 2019, 11, 39, 35770–35776; *Chem. Commun.*, 2018,54, 4453–4456). Also, could the authors highlight the novelty of the current work by comparing with the above papers?*

Response

Thank you for your valuable comments. Related works including those papers recommended by the reviewer have been listed in **Supplementary Table S2**.

For *ACS Appl. Mater. Interfaces* 2021, 13, 39, 46634–46643, DMAC was used to suppress water activity by increasing the number of hydrogen bonds and decreasing the number of free O-H bonds, as evidenced by the **redshift** of O-H vibrations in Raman spectra. However, in our work the function of DMAC is to increase the O-H bond energy *via* increasing electron density of water protons derived from the strong interaction between water and the DMAC. Importantly, O-H vibrations undergo **blueshift** in our hybrid electrolyte (HE). The difference between these two functions is exhibited in low-temperature performance. In the former design (*ACS Appl. Mater. Interfaces* 2021, 13, 39, 46634–46643), the number of hydrogen bonds increases at low temperature, therefore the lowest operation temperature for electrolyte is 0 °C. Importantly however, our HE works well at –40 °C. Different functions derive from salt concentration and the ratio of DMAC/H₂O.

The function of TMP in our work and that in Li-ion batteries (*J. Electrochem. Soc.*, 2001, 148, A1058, *ACS Appl. Mater. Interfaces*, 2019, 11, 39, 35770–35776 and *Chem. Commun.*, 2018, 54, 4453–4456) is similar in improving electrolyte safety. However, TMP has a negative impact on Li-ion batteries because of side reactions. Whilst TMP plays a positive role in our

HE, the electron-rich region of TMP, P=O group, assists to increase electron density of water protons and further strengthen O-H bonds.

In another report, *Angew. Chem. Int. Ed.*, 2019, 58, 2760-2764, is the first attempt to use TEP fire-retardant as a solvent for Zn anodes. However, a detailed discussion on solvation structure was not included. Significantly, Zn deposition in the reported electrolyte is a porous structure, however it is a close-packed morphology with (002) plane preferred crystal orientation in our HE. TMP together with DMAC alters the surface energy of Zn, and therefore leads to exposure of the (002) plane. This phenomenon has not been reported. Importantly, electrochemical performance is much superior.

Usually, non-aqueous solvents are added into aqueous electrolyte to regulate solvation structures. The main mechanisms for these strategies are the breaking of hydrogen between water molecules, restricting the activity of water in solvation sheaths, or excluding water molecules from the electric double layer (EDL), which aim to suppress water decomposition, lower the freezing point and boost dendrite-free Zn-plating. In our work, we demonstrate a new strategy to achieve these goals *via* introducing strong polar molecules, DMAC and TMP, to interact with water and therefore increase electron density of water protons and strengthen O-H bonds of water. These we evidenced *via* DFT, NMR and FT-IR findings. DMAC alters the surface energy for Zn and therefore guides (002) plane preferred orientation of Zn-deposition. This phenomenon has not been reported. Our electrolyte design principle is novel and distinguished from reported results.

We have in our R-MS, Page 6 line 1-11 and Page 17 line 6-24, included the comparison and discussion as follow:

'It was found that these peaks all undergo apparent blueshift in the spectra for HE, demonstrating that hydrogen bonds are weakened whilst O-H bonds strengthened²⁶. This result is totally different from the previous report, that adding DMAC into the aqueous electrolyte leads to redshift of O-H vibrations because of increasing number of hydrogen bonds. The ratio of DMAC/H₂O is the crucial factor that determines the hydrogen bond strengthened or weakened³⁰.'

*'A comparison of state-of-the-art electrolytes is listed in Supplementary **Table S2**. For non-aqueous electrolytes, the choices of compatible cathode materials are quite limited and the electrochemical properties, such as specific capacity and lifespan, of these cathodes are inferior compared with aqueous batteries^{17, 22}. As for those aqueous-based electrolytes, in which water accounts for > 50% by volume, the low-rate performance and high-temperature performance of batteries become the shortcomings^{26, 40, 41}, because the interactions between water and non-aqueous solvents are not sufficient to suppress the water activity. In our design, we emphasized the impact of water content on the specific capacity and stability of the NaV₃O₈ cathode and further demonstrated that the stability is highly related to the structural evolution of the cathode during battery operation. Upon comparison, the Zn||NaV₃O₈ battery in HE has advantages at a low cycling current density (0.1 A g⁻¹), and has superior temperature adaptability. By adding non-aqueous solvents to aqueous electrolytes, the improvement in Zn reversibility is quite general, however, the regular morphology of Zn plating is not that universal. Although (002) preferred orientation also occurs in some electrolyte formulas^{23, 26}, but the underlying mechanism has not been well explained. In our work, we confirm that DMAC and TMP could modify the surface energy of Zn and therefore guide the exposure of the (002) plane during Zn plating. DMAC³⁰ and TMP/TEP^{22, 24} have been reported in hybrid electrolytes, however, their functions in strengthening the O-H bonds and guiding the growth of the Zn (002) plane are not reported yet.'*

Supplementary Table S2. Comparative summary of state-of-art selected electrolytes for ZIBs.

	Electrolyte	Flammability (Y/N)	Preferred Zn plating orientation (Y/N)	CE for Zn plating-stripping	Oxidation voltage (vs. Zn ²⁺ /Zn)	Cathode	Cycle life	References
1	2 M Zn(OTf) ₂ + 7 M DEC aqueous electrolyte	N	N	Zn Cu cell, 99.24 %, 400 cycles, 1 mA cm ⁻² , 1 mAh cm ⁻²	N/A	V ₂ O ₅ ·nH ₂ O (mass loading N/A)	202 mAh g ⁻¹ with 72.9 % retention (2 A g ⁻¹), 5000 cycles (RT)	ACS Nano , 2022, 16, 9667–9678.
2	0.5 M Zn(OTf) ₂ in TMP/DMC	N	N	Zn SS cell, 99.15 %, 300 cycles, current and capacity N/A	2.25 V	VS ₂ , 4 mg cm ⁻²	107.86 mAh g ⁻¹ (100 mA g ⁻¹), 500 cycles (RT)	Adv. Mater. , 2019, 31, 1900668.
3	3 M Zn(CF ₃ SO ₃) ₂ in PC/H ₂ O (2:8)	N	N	N/A	N/A	NaV ₃ O ₈ ·1.5 H ₂ O, 8 mg cm ⁻²	168 mAh g ⁻¹ (0.2 A g ⁻¹) 400 cycles, -40 °C 229 mAh g ⁻¹ (5 A g ⁻¹) 1000 cycles, 30 °C	Adv. Funct. Mater. , 2022, 32, 2111714.
4	0.5 M Zn(CF ₃ SO ₃) ₂ in TEP/H ₂ O (7:3)	N	N	Zn SS cell, 93.71 %, 100 cycles, current and capacity N/A	2.25 V	KCuHCF, 1 mg cm ⁻²	~50 mAh g ⁻¹ , 74 % retention, 1000 cycles, average CE 97.66 %	Angew. Chem. Int. Ed. , 2019, 58, 2760–2764.
5	4 M Zn(TFSI) ₂ + 4 M P ₄₄₄ (201)-TFSI in H ₂ O	N/A	N	Zn Cu cell, ~99 %, 1.17 mA cm ⁻² , 1.17 mAh cm ⁻² , 20% Zn utilization, 16 cycles	2.42 V	Na ₂ V ₆ O ₁₆ ·1.6 H ₂ O, 1 mg cm ⁻²	~120 mAh g ⁻¹ , 1900 cycles (300 mA g ⁻¹) (RT)	Angew. Chem. Int. Ed. , 2021, 60, 12438–12445.
6	1 M Zn(CF ₃ SO ₃) ₂ in PC/H ₂ O (5:5)	N	Y	Zn Cu cell, 99.93 %, 1 mA cm ⁻² , 0.5 mAh cm ⁻² , 500 cycles	~ 2.5 V	PANI, 1.5-2 mg cm ⁻²	78 mAh g ⁻¹ (0.2 A g ⁻¹) 200 cycles, -20 °C 95 mAh g ⁻¹ (0.2 A g ⁻¹) 200 cycles, 50 °C	J. Am. Chem. Soc. , 2022, 144, 7160–7170.
7	3 M ZnSO ₄ + 10 mM α-cyclodextrin aqueous electrolyte	N	Y	Zn Cu cell, ~99.9 %, 1 mA cm ⁻² , 1 mAh cm ⁻² , 600 cycles	~ 2.5 V	V ₂ O ₅ , 6.4 mg cm ⁻²	~180 mAh g ⁻¹ , 200 cycles (1 A g ⁻¹) (RT)	J. Am. Chem. Soc. , 2022, 144, 11129–11137.
8	2 M ZnSO ₄ and 0.0085 M La(NO ₃) ₃ aqueous electrolyte	N	Y	Zn Ti cell, ~99.9 %, 2 mA cm ⁻² , 1 mAh cm ⁻² , 2200 cycles	N/A	VS ₂ , 8 mg cm ⁻² (N/P ratio 4:3)	120 mAh g ⁻¹ (100 mA g ⁻¹), 100 cycles (RT)	Nat. Commun. , 2022, 13, 3252.

9	1 mol/kg Zn(TFSI) ₂ + 20 mol/kg LiTFSI aqueous electrolyte	N	N	Zn Pt (three electrode cell), 99.7 %	N/A	LiMn ₂ O ₄ , 2.4 mAh cm ⁻²	26 mAh g ⁻¹ (4 C), 4000 cycles (RT)	Nat. Mater. , 2018, 17, 543–549.
10	2 M ZnSO ₄ in NMP/H ₂ O (5:5)	N/A	Y	Zn Cu cell, ~99.7 %, 1 mA cm ⁻² , 0.5 mAh cm ⁻² , 1000 cycles	N/A	VS ₂ , 5.3 mg cm ⁻²	125 mAh g ⁻¹ (1 A g ⁻¹), 2000 cycles (RT)	Adv. Energy Mater. , 2022, 12, 2103231.
11	3 M ZnSO ₄ + 0.5 M glycine aqueous electrolyte	N	Y	Zn Cu cell, 99.68 %, 2 mA cm ⁻² , 2 mAh cm ⁻² , 650 cycles	N/A	NH ₄ V ₄ O ₁₀ (mass loading N/A)	220 mAh g ⁻¹ (5 A g ⁻¹), 3000 cycles (RT)	ACS Nano , 2022, DOI: 10.1021/acsnano.2c09317
12	1 M ZnAc ₂ + 4 M NH ₄ I aqueous electrolyte	N	N	Zn Cu cell, 99.8 %, 1 mA cm ⁻² , 1 mAh cm ⁻² , 100 cycles	N/A	I ₂ (mass loading N/A)	~1 mAh cm ⁻² , 200 cycles	J. Am. Chem. Soc. , 2022, 144, 18435–18443
13	1.3 M ZnCl ₂ in H ₂ O/DMSO (volume ratio of H ₂ O/DMSO = 4.3:1)	N/A	N	Zn Cu cell, 99.5 %, 1 mA cm ⁻² , 0.5 mAh cm ⁻² , 400 cycles	~2.2 V	MnO ₂ (mass loading N/A)	150 mAh g ⁻¹ (8 C), 500 cycles (RT)	J. Am. Chem. Soc. , 2020, 142, 21404–21409.
14	BMITFSI:Zn(TFSI) ₂ (water/ionic liquid mass 20 %)	N	N	Zn Cu cell, 99.27 %, 1 mA cm ⁻² , 0.5 mAh cm ⁻² , 400 cycles	~2.5 V	PANI, 0.8-0.95 mg cm ⁻²	18 mAh g ⁻¹ (1 A g ⁻¹), 1000 cycles (RT)	ACS Energy Lett. , 2023, 8, 608–618.
15	2 M ZnSO ₄ in water/EG (volume ratio of EG 40 %)	N	N	Zn Ti cell, ~98 %, 2 mA cm ⁻² , 1 mAh cm ⁻² , 120 cycles	~2.8 V	PANI/V ₂ O ₅ , 2.5-3.0 mg cm ⁻²	~45 mAh g ⁻¹ (0.2 A g ⁻¹), 6500 cycles, -20 °C ~60 mAh g ⁻¹ (5 A g ⁻¹), 50000 cycles, 20 °C	Energy Environ. Sci. , 2020, 13, 3527–3535.
16	1 M ZnTFMS/DMF	Y	N	Zn SS cell, ~99.8 %, 1 mA cm ⁻² , 1 mAh cm ⁻² , 200 cycles	~2.4 V	PQMCT (mass loading N/A)	~22 mAh g ⁻¹ (0.2 A g ⁻¹), 1 cycle, -70 °C ~180 mAh g ⁻¹ (2 A g ⁻¹), 1 cycle, 150 °C	Angew. Chem. Int. Ed. , 2020, 59, 14577 – 14583
17	7.5 mol/Kg ZnCl ₂ -based aqueous electrolyte	N/A	N	Zn Cu cell, ~99 %, 0.2 mA cm ⁻² , 0.2 mAh cm ⁻² , 50 cycles	~1.9 V	PANI, (mass loading N/A)	~75 mAh g ⁻¹ (0.2 A g ⁻¹), 2000 cycles, -70 °C	Nat Commun. , 2022, 11, 4463.
18	4 M Zn(BF ₄) ₂ aqueous electrolyte	N	N	Zn SS cell, ~95 %, 0.5 mA cm ⁻² , 0.5 mAh, 1 cycle	~2 V	TCBQ2, 1-2 mg	~95 mAh g ⁻¹ (1 C), 1000 cycles, -30 °C ~80 mAh g ⁻¹ (0.1 C), 50 cycles, -30 °C	J. Mater. Chem. A , 2021, 9, 7042–7047.

19	2 M Zn(OTf) ₂ in water/DMC (volume ratio 4:1)	N	N	Zn Ti cell, ~99.8 %, 1 mA cm ⁻² , 1 mAh cm ⁻² , 200 cycles	N/A	V ₂ O ₅ , 2 mg cm ⁻²	~380 mAh g ⁻¹ (2 A g ⁻¹) 1000 cycles, RT	Chem. Sci. , 2021, 12, 5843–5852.
20	2.5 M Zn(NO ₃) ₂ +13 M LiNO ₃ in DMA/H ₂ O	N/A	N	N/A	~1.9 V	LiMn ₂ O ₄ , 3 mg cm ⁻²	95 mAh g ⁻¹ (1 C), 200 cycles (RT)	ACS Appl. Mater. Interfaces , 2021, 13, 39, 46634–46643.
21	1 M Zn(OTf) ₂ in DMAC/TMP/H ₂ O (volume ratio 5:2:3)	N	Y	Zn Cu cell, 99.5 %, 1 mA cm ⁻² , 1 mAh cm ⁻² , 2000 cycles	~2.25 V	NaV ₃ O ₈ ~6 mg cm ⁻² for RT testing; ~0.6 mg cm ⁻² for high-low temperature testing	~264 mAh g ⁻¹ (0.1 A g ⁻¹) 700 cycles, (RT); ~190 mAh g ⁻¹ (0.5 A g ⁻¹) 3000 cycles, (RT); ~100 mAh g ⁻¹ (0.1 A g ⁻¹) 1800 cycles, -40 °C; ~200 mAh g ⁻¹ (5 A g ⁻¹) 1400 cycles (retention ~60 %), 70 °C;	This work

Comment 1-3

Is 5:2:3 the optimized ratio for DMAC/TMP/H₂O solvent? For Zn salts, is the proposed ternary solvent compatible with ZnSO₄ and Zn(TFSI)₂? The former is the most commonly used Zn salt (and cheap), while the latter has been found to enable promising Zn²⁺ storage properties for V-based and Mn-based cathodes.

Response

5:2:3 is the optimal ratio for DMAC/TMP/H₂O for electrochemical performance of electrodes and safety.

As is shown in **Supplementary Fig. S1** the (002) preferred orientation for Zn deposition is achieved when the volume ratio of H₂O in the DMAC/H₂O mixture is in the range of 20 to 40 %. The ratio of H₂O has a significant impact on the specific capacity and stability of NaV₃O₈ cathode, and 30 % water is the ‘best’ option. With 20% water, the specific capacity for NaV₃O₈ is (relatively) low, and with 40 % water, the specific capacity is greater for the first several cycles, however the rate of decay is faster (**Supplementary Fig. S2**).

A drawback is that the DMAC/H₂O mixture is flammable. Fire retardant TMP is used to partly replace DMAC to ensure non-flammability of the electrolyte; however, this needs to be minimized to maintain low viscosity. The data of **Supplementary Fig. S3** confirm the minimum TMP is *ca.* 30 % in the mixture of DMAC/TMP, therefore, the ratio of TMP in the DMTC/TMP/H₂O is 70 % × 30 % = 21 %. Taking into consideration Zn deposition morphology, capacity of electrode, and non-flammability of electrolyte, we optimized electrolyte with DMAC/TMP/H₂O of 5 (~70 % × 70 %):2 (~70 % × 30 %): 3 (30 %).

This optimized electrolyte cannot be ignited. Coulombic efficiency (CE) for Zn plating/stripping reaches an average value of CE = 99.5 %, and dendrite-free Zn deposition is exhibited. The specific capacity of cathode is *ca.* 200 mAh g⁻¹ at a charge/discharge current density 0.5 A g⁻¹, with the cycling life of the full cell significantly increased to 3000 cycles.

Supplementary Fig. S1. Morphology evolution for Zn deposited in 1 M Zn(OTf)₂ solution. Solvent is a DMAC/H₂O mixture with varying percent volume H₂O.

Supplementary Fig. S2. Cyclical performance for Zn||NaV₃O₈ cells in 1 M Zn(OTf)₂ solution. Solvent is DMAC/H₂O mixture with varying percent volume H₂O.

Supplementary Fig. S3. Digital images of ignition test for 1 M Zn(OTf)₂ solution. The Solvent is (a) DMAC, (b) DMAC/TMP (8:2 by volume), (c) DMAC/TMP (7:3 by volume), (d) DMAC/H₂O (5:3 by volume) and (e) DMAC/TMP /H₂O (5:2:3 by volume).

ZnSO₄ is insoluble and low cost inorganic salt, and Zn(ClO₄)₂ works well in hybrid solvent. The ternary solvent is compatible with Zn(TFSI)₂. **Supplementary Fig. S28** compares performance for the NaV₃O₈ cathode and Zn anode in the hybrid and aqueous, electrolytes using, respectively, Zn(TFSI)₂ and ZnClO₄ as salt. Advantages with hybrid electrolytes are significant both in boosting cycling life of full cells and in protecting the Zn anode.

Supplementary Fig. S28. Comparison of cycling performance for Zn||NaV₃O₈ cells and CE for Zn plating/stripping in hybrid and aqueous, electrolyte. (a) (b) 1 M Zn(TFSI)₂ used as salt and (c) (d) 1 M Zn(ClO₄)₂ used as salt. In hybrid electrolyte, solvent is DMAC/TMP/H₂O in a volume ratio 5:2:3.

In response to this comment we have in our R-MS, Page 5 line18-27 and Page 14 line 23-27, included additional clarifying text, namely:

‘Electrolyte formulation was based on three principles, namely: 1) dendrite-free deposition of Zn; 2) good compatibility with NaV₃O₈ cathode, and; 3) non-flammability. It was observed that the Zn deposition orientates the (002) plane when the volume percentage of water in DMAC/H₂O mixture ranged from 20 to 40 % (Supplementary Fig. S1). Water content has a highly significant impact on cathode performance. The NaV₃O₈ electrode exhibited a satisfactory capacity and cyclic stability when the ratio of H₂O was 30 % (Supplementary Fig. S2). TMP is added to ensure the formulated electrolyte is nonflammable, and usage is around 30 % in the mixture of DMAC/TMP (Supplementary Fig. S3). Therefore, the volume ratio for DMAC/TMP/H₂O was determined to be 5:2:3. 1 M Zn(OTf)₂ dissolved in this mixture is denoted HE whilst 1 M Zn(OTf)₂ aqueous solution, AE.’

‘DMAC/TMP/H₂O hybrid solvent in a volume ratio of 5:2:3 is compatible with salts including zinc trifluoromethanesulfonate (Zn(TFSI)₂) and Zn(ClO₄)₂. In hybrid electrolytes using Zn(TFSI)₂ or ZnClO₄ as salt, the NaV₃O₈ cathode maintains specific capacity without decay for > 1200 cycles, and the CE for Zn plating/stripping is boosted compared with that in corresponding aqueous electrolyte (Supplementary Fig. S28).’

Comment 1-4

(002) preferred orientation is proposed to be a key factor in deposited Zn. However, XRD results were not provided.

Response

Thank you for your comment. In response to address this comment we have in our R-SI included XRD patterns for Zn deposited in HE and AE as **Supplementary Fig. S6**.

It is seen in the figure that when Zn^{2+} is deposited in HE the intensity of the (002) reflection significantly increases compared with that for AE.

Supplementary Fig. S6. The XRD patterns for commercial Zn-foil, Zn deposited in AE and in HE.

Comment 1-5

Are there any Na residuals in the V_2O_5 sample? Mixing NaCl and V_2O_5 usually results in the precipitation of NaV_3O_8 (for instance, *Nat. Commun.*, 2018, 9:1656; *ACS Appl. Energy Mater.* 2018, 1, 6401), but V_2O_5 was obtained in the current work. Could the authors comment on this?

Response

To identify the material obtained we double-checked the XRD pattern, synchrotron-based X-ray powder diffraction (XRPD) pattern and TEM images, and found that the powders are NaV_3O_8 and not in fact V_2O_5 .

The material is synthesized following the method reported in two (2) published papers, *Nano Energy*, 2016, 22: 583–593 and *Nano Energy*, 2019, 60: 752-759, where the product was identified as V_2O_5 , with a nanobelt morphology orientating the [010] direction. We did not question this when the material exhibited the same morphology and a clear XRD pattern for V_2O_5 in our initial synthesis.

However, in response to this comment of Reviewer #1 we applied high-resolution TEM on the nanobelts and found the lattice fringe did not match the lattice parameters for V_2O_5 but is consistent with that for NaV_3O_8 .

We realized that stirring speed significantly affects formation of NaV_3O_8 during synthesis. At a high stirring speed of 400 rpm, the growth rate of NaV_3O_8 grains is low and a longer time, *ca.* 72 h needed to complete reaction. Whilst at a low stirring speed of 200 rpm, grains grow faster and nanobelts appear, even at 50 h. However, there are residual V_2O_5 powders even following a 72 h reaction because of insufficient stirring. In the XRD pattern previously provided (see below) the strong reflections of V_2O_5 originate from these residual V_2O_5 powders. Because of low crystallinity the diffraction peaks for NaV_3O_8 are weaker than those for V_2O_5 , affecting visibility.

Fig. R1. XRD pattern for powders stirred at low speed for 72 h.

Once the reaction is completed, the XRD pattern for the obtained material is consistent with that for NaV_3O_8 (**Supplementary Fig. S18c**). Elemental mapping images (**Supplementary Fig. S18d**) confirm the existence of Na in the nanobelt. A 0.77 nm interplanar spacing was observed in the high-resolution TEM image corresponding to the (001) plane for NaV_3O_8 . The lattice distance agrees well with the 250 °C heated NaV_3O_8 instead of 80 °C heated because the interlayer water was extracted in the high-vacuum sample chamber for TEM.

We sincerely appreciate the reviewer pointing out this mistake. The relevant information has been corrected in the R-MS.

Supplementary Fig. S18. (a) TEM image for NaV_3O_8 nanobelt. (b) The high-resolution TEM image. (c) XRD pattern at 80 °C heated and 250 °C heated NaV_3O_8 . (d) TEM elemental mapping image for NaV_3O_8 nanobelt.

Comment 1-6

Pristine V_2O_5 has predominant (001) diffraction peak without noticeable (020) reflection. However, *in situ* XRD data only show the changes of (020) reflection.

Response

We have double-checked data for *in operando* synchrotron-based XRPD. We confirm no residual V_2O_5 in the electrode, and the reflection assigned previously to the (020) plane of V_2O_5 should be the (204) plane of NaV_3O_8 .

Supplementary Fig. S25a presents the synchrotron powder diffraction pattern for the electrode, and is consistent with the XRD pattern for NaV_3O_8 (Supplementary Fig. S18c). However, when the NaV_3O_8 electrode is assembled in the battery, its (001) reflection becomes too weak to monitor changes during charge/discharge. Significantly, the (204) reflection remains clear, therefore, we conclude that it is the ‘best’ option for observation of structural evolution. In addition, the shielding effect of Zn/Ti foils and pressure applied during battery assembly might account for this phenomenon.

Supplementary Fig. S25. (a) XRPD pattern for NaV_3O_8 electrode. (b) XRPD pattern for initial state of *in operando* $\text{Zn}||\text{NaV}_3\text{O}_8$ cell.

Comment 1-7

The high reversibility of V_2O_5 in HE is attributed to good structural stability (compared with AE). Could the authors provide XRD data before and after cycling in AE and HE to support this statement?

Response

Because of an unsatisfactory signal-to-noise ratio of laboratory-based XRD (**Fig. R2**), we applied synchrotron-based XRPD on the NaV_3O_8 electrode following cycling.

In HE the structure for NaV_3O_8 is maintained during cycling and no side product(s) was found (**Fig. 5a**). However, in AE $Zn_4SO_4(OH)_6 \cdot 4H_2O$ formed and the (001) reflection weakened and all but disappeared following 200 cycles (**Supplementary Fig. S26**).

Fig. R2. Laboratory-based XRD pattern for pristine NaV_3O_8 electrode.

Fig. 5. (a) Synchrotron-based XRPD pattern for NaV_3O_8 electrode following cycling in HE.

Supplementary Fig. S26. Synchrotron-based XRPD pattern for NaV_3O_8 electrode following cycling in AE.

In response to this comment, we have in our R-MS, Page 14 line 17-21, included the following additional text:

'Fig. 5a presents the XRPD pattern for NaV_3O_8 electrode following cycling in HE for 50, 100 and 200 cycles, evidencing that the structure of NaV_3O_8 is maintained during cycling and no side product(s) is found. However, in AE, $\text{Zn}_4\text{SO}_4(\text{OH})_6 \cdot 4\text{H}_2\text{O}$ formed and the (001) reflection weakened and all but disappeared following 200 cycles (Supplementary Fig. S26).'

Comment 1-8

One of the merits of HE is wider ESW. But the redox reactions of V-based materials are relatively low. Could the authors demonstrate the performance of other cathode materials with high operating voltage?

Response

To determine the compatibility of HE with the high-voltage cathode, zinc hexacyanoferrates ($\text{Zn}_3(\text{Fe}(\text{CN})_6)_2$), was selected with an operation voltage of 1.8 to 1.6 V when combined with Zn anode. The initial specific capacity for $\text{Zn}||\text{Zn}_3(\text{Fe}(\text{CN})_6)_2$ cells is *ca.* 60 mAh g^{-1} in both electrolytes (Supplementary Fig. S30a), however it decreases rapidly in AE accompanied by voltage decay (Supplementary Figs. S30b-c). The output voltage for $\text{Zn}_3(\text{Fe}(\text{CN})_6)_2$ reduces to 1.6 to 1.4 V following 150 cycles in AE (Supplementary Fig. S30d). This apparent disadvantage is attributed to material degradation and less stability of H_2O at high voltage during charging causing low CE. In contrast, the $\text{Zn}||\text{Zn}_3(\text{Fe}(\text{CN})_6)_2$ cell with HE exhibits better

stability and high-voltage character is well maintained even following 150 cycles, evidencing that HE is compatible with high-voltage cathode.

Supplementary Fig. S30. (a) Cycling performance for Zn||Zn₃(Fe(CN)₆)₂ cells tested at 50 mA g⁻¹ in AE and HE. (b-d) Voltage profiles for, respectively, 5th, 20th and 150th cycles.

In response to this comment, we have in our R-MS, Page 15 line 7-18, included the following clarifying text:

‘To determine the compatibility of HE with high-voltage cathode, zinc hexacyanoferrates (Zn₃(Fe(CN)₆)₂), was selected that exhibits an operation voltage of 1.8 to 1.6 V when combined with the Zn anode. The initial specific capacity for Zn||Zn₃(Fe(CN)₆)₂ cells is ca. 60 mAh g⁻¹ in both electrolytes (Supplementary Fig. S30a). However, it decreases rapidly in AE accompanied by voltage decay (Supplementary Figs. S30b-c). The output voltage for Zn₃(Fe(CN)₆)₂ reduces to 1.6 to 1.4 V following 150 cycles in AE (Supplementary Fig. S30d). This is attributed to material degradation and less stability of H₂O at high voltage during charging, that causes relatively low CE. In contrast, the Zn||Zn₃(Fe(CN)₆)₂ cell exhibits better stability in HE and the high-voltage character is well maintained following after 150 cycles, evidencing that HE is compatible with the high-voltage cathode.’

Comment 1-9

What is the ionic conductivity of the HE (compared with AE)? Rate performance is also important for batteries. Moreover, it is recommended to provide a comparison table including the electrochemical performance of the current work and reported papers.

Response

The ionic conductivity of AE and HE at room temperature is, respectively, 41.9 and 9.0 mS cm⁻¹. Rate performance for Zn||NaV₃O₈ cells is provided in Fig. 4a.

A comparative summary of our findings with those reported is now in our R-SI as **Supplementary Table S2**.

In response to this comment, we have in our R-MS, Page 15 line 2-6, Page 12 line 27-Page 13 line 11 and Page 17 line 8-25, included additional explanatory text, namely:

*‘The ionic conductivity and viscosity of AE and HE are given in **Supplementary Table S1**. AE has advantages in both conductivity and viscosity at temperature > 0 °C. A Walden plot can be drawn via calculating molar conductivity and viscosity, in which it is evidenced that $\text{Zn}(\text{OTf})_2$ salt is well-dissociated in these two electrolytes.’*

*‘The rate performance for NaV_3O_8 electrode was tested in AE and HE, at a high active material mass loading of 9.3 mg cm^{-2} and low areal capacity ratio of negative to positive electrodes, N/P ratio=5. Because of high ionic conductivity of AE, the $\text{Zn}||\text{NaV}_3\text{O}_8$ cell exhibits better rate performance, with ca. 80 mAh g^{-1} at 5 A g^{-1} , as is seen in **Fig. 4a**. The $\text{Zn}||\text{NaV}_3\text{O}_8$ cell using HE does not exhibit an advantage in specific capacity, especially when the applied current density increases.’*

*‘Although HE has not a superiority in rate performance, it significantly improves stability of $\text{Zn}||\text{NaV}_3\text{O}_8$ cells. At a low current density of 0.1 A g^{-1} the specific capacity for the aqueous $\text{Zn}||\text{NaV}_3\text{O}_8$ cell faded significantly within 200 cycles, whilst in HE, the $\text{Zn}||\text{NaV}_3\text{O}_8$ cell maintained a specific capacity of 164 mAh g^{-1} following 700 cycles (**Supplementary Fig. S22**). At a current density of 0.5 A g^{-1} , in AE, the NaV_3O_8 cathode exhibited a (rapid) increase in specific capacity in the first 20 cycles from 170 to 231 mAh g^{-1} , and an apparent decrease with subsequent cycling, only to fail completely in following 400 cycles. Corresponding charge-discharge curves showed that voltage decayed highly significantly following 300 cycles (**Supplementary Fig. S23**).’*

*‘A comparison of state-of-the-art electrolytes is listed in **Supplementary Table S2**. For non-aqueous electrolytes, the choices of compatible cathode materials are quite limited and the electrochemical properties, such as specific capacity and lifespan, of these cathodes are inferior compared with aqueous batteries^{17, 22}. As for those aqueous-based electrolytes, in which water accounts for $> 50\%$ by volume, the low-rate performance and high-temperature performance of batteries become the shortcomings^{26, 40, 41}, because the interactions between water and non-aqueous solvents are not sufficient to suppress the water activity. In our design, we emphasized the impact of water content on the specific capacity and stability of the NaV_3O_8 cathode and further demonstrated that the stability is highly related to the structural evolution of the cathode during battery operation. Upon comparison, the $\text{Zn}||\text{NaV}_3\text{O}_8$ battery in HE has advantages at a low cycling current density (0.1 A g^{-1}), and has superior temperature adaptability. By adding non-aqueous solvents to aqueous electrolytes, the improvement in Zn reversibility is quite general, however, the regular morphology of Zn plating is not that universal. Although (002) preferred orientation also occurs in some electrolyte formulas^{23, 26}, but the underlying mechanism has not been well explained. In our work, we confirm that DMAC and TMP could modify the surface energy of Zn and therefore guide the exposure of the (002) plane during Zn plating. DMAC³⁰ and TMP/TEP^{22, 24} have been reported in hybrid electrolytes, however, their functions in strengthening the O-H bonds and guiding the growth of the Zn (002) plane are not reported yet.’*

Fig. 4a. Rate performance for Zn||NaV₃O₈ cells tested in AE and HE. Mass loading for NaV₃O₈ is 9.3 mg cm⁻².

Comment 1-10

The experimental details should be provided, such as the grade of Zn anode (thickness, purity...), supplier of battery separators, and mass loading for V₂O₅ cathode.

Response

We agree with this comment of Reviewer #1.

In response to this request we have in our R-SI, Page 3 line 15-22, included the following detailed information, namely:

'Zn-foil (99.99 %, 100 μm) was used as the anode. For batteries using aqueous electrolyte (AE) a glass-fibre membrane (740 μm) was used as the separator, and for those with hybrid electrolyte (HE) or non-aqueous electrolytes, nylon 66 membrane (130 μm) was used. The glass-fibre membrane was purchased from Filtech Pty Ltd, whilst the nylon 66 membrane from Shanghai Xinya Purification Equipment Co., Ltd. The mass loading for NaV₃O₈ cathode ranged from 0.6 to 9.3 mg cm⁻², the exact mass is given in the legend for each figure.'

Response to Reviewer #2

Reviewer's Remarks to Authors

Through this good scientific paper, the authors present a battery based on a new aqueous electrolyte, mainly by playing on the control of the O-H bond. Despite the interest I had in reading the scientific paper, some weaknesses are present here and do not allow a firm conclusion. 2/5 of the identified weaknesses are significant to accept the present work (comments (iii) and (iv)).

Response

We thank Reviewer #2 for his/her valuable comments.

Comment 2-(i)

Lack of standard measurement of formulated electrolytes. It is now recognized that impurities in formulated electrolytes in batteries influence performance and, at the same time, confound measurement. Thus the characterization of electrolytes (as well as electrodes) must be characterized with rigor and detail. In the present case is missing, to my knowledge:

1. Thermal measurements: DSC, TGA
2. Ionic conductivity and viscosity (Walden Plot).

Response

To reduce impurity, chemicals used to formulate the electrolytes were of the highest commercial purity available, that is, 99.8 % for DMAC, $\geq 99\%$ for TMP and 98 % for Zn(OTf)₂. The water was deionized and purified in Laboratory deionization with a purity measuring $10^6 \Omega$.

To determine the operating temperature for AE and HE, differential scanning calorimetry (DSC) and thermogravimetric analysis (TGA) were conducted. For AE, there is an exothermic peak at *ca.* -29°C corresponding to crystallization of electrolyte, and an endothermic peak at -4°C corresponding to melting. In contrast, no exothermic/endothermic peaks were observed for HE despite the temperature cooling to -80°C (**Supplementary Fig. S32a**). TGA curves showed that AE loses mass more rapidly than HE because of fast H₂O volatilization. In HE strong interactions between H₂O and DMAC/TMP reduce volatilization, increasing stability for HE at high temperatures (**Supplementary Fig. S32b**).

Fig. S32. (a) DSC curves for water, AE and HE. (b) TGA curves for AE and HE under N₂ atmosphere using a heating rate of $20^\circ\text{C min}^{-1}$.

Walden's rule confirms that the product of molar conductivity (A) of a liquid solution and its viscosity (η) is a constant at a given temperature, as expressed by:

$$A \eta = C = \text{Constant} \quad (1)$$

$$\log A = \log C + \log \eta^{-1} \quad (2)$$

The data for dilute aqueous KCl solution, a solution in which all ions are dissociated, can be used as an 'ideal' Walden line. In **Supplementary Fig. S33**, both the Walden plot for AE and HE are close to this ideal line, evidencing that Zn(OTf)₂ salt is well-dissociated in these two electrolytes.

Supplementary Fig. S33. Walden plot of temperature-dependent conductivity and viscosity for AE and HE.

In response to this request we have in our R-SI, Page 15 line 17- Page 16 line 6, included the following detailed discussion:

'To investigate the operating temperature of AE and HE, differential scanning calorimetry (DSC) and thermogravimetric analysis (TGA) were conducted. For AE, there is an exothermic peak at around $-29\text{ }^{\circ}\text{C}$ corresponding to the crystallization of electrolyte and an endothermic peak at $-4\text{ }^{\circ}\text{C}$ corresponding to the melting point. In contrast, no exothermic/endothermic peaks are observed for HE even with the temperature cooling down to $-80\text{ }^{\circ}\text{C}$ (Supplementary Fig. S32). The TGA curves show that AE loses weight faster than HE because of fast H_2O volatilization. While in HE, the strong interactions between H_2O and DMAC/TMP effectively reduce volatilization effects, increasing the stability of HE at high temperatures. The ionic conductivity and viscosity of AE and HE are given in Supplementary Table S1. AE has advantages in both conductivity and viscosity at temperatures $> 0\text{ }^{\circ}\text{C}$. A Walden plot can be drawn via calculating molar conductivity and viscosity, in which it is evidenced that $\text{Zn}(\text{OTf})_2$ salt is well-dissociated in these two electrolytes (Supplementary Fig. S33).'

Comment 2-(ii)

The cell voltage measurement is strongly related to the ESW. In this case, the effect on the positive electrode is unambiguously observed. However, no measurements at the negative electrode are made to identify the ESW. This is a significant shortcoming.

1. There is a need to study ESW at both the positive and negative electrodes. The authors should ideally carry out this study in LSV via ultramicroelectrodes to avoid the ohmic drop. But already providing a measurement of the limit in negative potential is necessary.

Response

Thank you for your valuable comments.

We therefore conducted a three-electrode LSV using Pt ultramicroelectrode with a diameter of $10\text{ }\mu\text{m}$ as working electrode, Pt plate electrode as counter electrode and Ag/AgCl electrode as reference electrode. The ESW for HE was measured by linear sweep voltammetry (LSV), that was stable up to $2.11\text{ V vs. Ag/AgCl}$ at a scanning rate of 1 mV s^{-1} . Importantly, this is greater than that for AE of 1.83 V , confirming greater stability for water molecules in HE than

in AE (**Fig. 1e**). The reduction of Zn^{2+} in AE and HE occurs at (almost) the same potential of -1.06 V, with the undifferentiated current response, evidencing that the overpotential needed for Zn deposition is the same as for AE and HE.

In response to this comment, we have in our R-MS, Page 6 line 31- Page 7 line 3, included the following discussion:

*'The ESW of HE is measured by linear sweep voltammetry (LSV), and it is stable up to 2.11 V versus Ag/AgCl at a scanning rate of 1 mV S^{-1} , obviously higher than that of AE (1.83 V), confirming the higher stability of water molecules in HE than that in AE (**Fig. 1 e**). The reduction of Zn^{2+} in AE and HE occurs at almost the same potential, -1.06 V, with the undifferentiated current response, indicating that the overpotential needed for Zn deposition is the same for the AE and HE.'*

Fig. 1. (e) LSV profiles for Pt ultramicroelectrode in AE and HE at a scan rate 1 mV s^{-1} .

2. In figure S22, the window of the GCD measurement is not convincing for a two-electrode system; the authors vary the voltage from 0.3 to 1.5. The value of 0.3 deserves justification.

Response

Thank you for your suggestion. A three-electrode CV was used to determine reduction and oxidation during charge/discharge of the $\text{Zn}||\text{NaV}_3\text{O}_8$ cell. **Supplementary Fig. S19** shows that no reduction occurs at voltage < 0.3 V (vs. Zn^{2+}/Zn) and no oxidation at a voltage > 1.3 V (vs. Zn^{2+}/Zn). Therefore, discharging cut-off at 0.3 V and charging ending at 1.6 V is sufficient for the $\text{Zn}||\text{NaV}_3\text{O}_8$ cell to complete electrochemical reaction(s). A 0.3 V as cut-off voltage for discharging was used in reported examples including, *ACS Nano*, 2020, 14: 6752–6760 and *Nat. Commun*, 2018, 9: 1656.

Supplementary Fig. S19. 3-electrode CV curve for NaV₃O₈ electrode scanned at a voltage between 0.1 to 1.6 V vs. Zn²⁺/Zn.

Comment 2-(iii)

This paper does not present performances measurements. Today, it seems necessary that device metrics be constructed with reproducibility and comparison (see doi.org/10.1002/batt.202100154 AND <https://doi.org/10.1038/s41467-022-29257-w>). In this respect, plotting a Ragone Plot (normalized by mass and then by volume) seems essential to conclude the interest of the electrolyte proposed here. To my knowledge, the present article does not provide energy density and maximum power values. This is one of the major criticisms of the present work: NO performances measurement allows us to conclude on the real interest of the work:

- 1 Lack of Ragon Plot.
- 2 Absence of a measure of Energy Density.
- 3 Lack of a Power Max.
- 4 Absence of different C-rate measurements.

Response

Thank you for your valuable comments. To answer this concern, we assembled Zn||NaV₃O₈ cells in which the mass loading for cathode active material was 9.3 mg cm⁻², the areal capacity ratio of negative to positive electrodes (N/P ratio) was controlled at 5 and, the electrolyte limited to 60 μL for HE and 120 μL for AE. Because of high ionic conductivity of AE, the Zn||NaV₃O₈ cell exhibited better rate performance, with *ca.* 80 mAh g⁻¹ at 5 A g⁻¹, as is seen in **Fig. 4a**. However, the Zn||NaV₃O₈ cell using HE did not exhibit any advantage in specific capacity, especially when the applied current density increased. As a result, the cell in AE exhibited a superior energy density of 65 Wh kg⁻¹, and greater power energy density of 727 W kg⁻¹, based on the mass of the cathode and anode. In HE the energy density of the cell is 56.5 Wh kg⁻¹ and the power maximum 500 W kg⁻¹.

It is widely acknowledged that the total mass of the battery is determined by technical parameters of electrode preparation and battery assembly, together with material and thickness

of the separator and packaging. The greatest mass loading of cathode active material we can achieve in our Laboratory is 9.3 mg cm^{-2} and, the lowest N/P ratio is 5. We used a nylon 66 membrane in the battery with HE as electrolyte. Nylon 66 is significantly thinner ($130 \text{ }\mu\text{m}$) and lighter (3.4 mg cm^{-2}) than widely used glass fibre separator of $740 \text{ }\mu\text{m}$ thickness and 13.7 mg cm^{-2} in mass, and therefore reduces the amount of electrolyte by 50 %. However, there remains room to reduce the mass of inert materials further, we are not able however to optimize all in a short time-frame.

The Ragone plot of **Fig. 4b** provides a comparison of electrochemical performance under different current densities in AE and HE, based on optimised cathode and anode electrodes. Although HE is not superior in rate performance, it significantly improves stability of $\text{Zn}||\text{NaV}_3\text{O}_8$ extending the lifespan of the cell to 700 cycles at a current density of 0.1 A g^{-1} (**Supplementary Fig. S22**) and 3000 cycles at 0.5 A g^{-1} (**Fig. 4c**).

In response to this comment we have in our R-MS, Page 12 line27-Page 13 line 6, included discussion as follows:

*‘The rate performance for the NaV_3O_8 electrode was tested in AE and HE at a high active material mass loading of 9.3 mg cm^{-2} and low areal capacity ratio of negative to positive electrodes (N/P ratio = 5). Because of high ionic conductivity of AE, the $\text{Zn}||\text{NaV}_3\text{O}_8$ cell exhibits a better rate performance with ca. 80 mAh g^{-1} at 5 A g^{-1} , as is seen in **Fig. 4a**. However, the $\text{Zn}||\text{NaV}_3\text{O}_8$ cell using HE does not exhibit any advantage in specific capacity, especially when applied current density increases. The energy and power, density for $\text{Zn}||\text{NaV}_3\text{O}_8$ batteries is given, together with computed the mass of the cathode and anode, **Fig. 4b**. In the Ragone plot, battery in AE exhibits a superior energy density of 65 Wh kg^{-1} and greater power energy density of 727 W kg^{-1} . In HE the energy density of the cell is 56.5 Wh kg^{-1} and the power maximum 500 W kg^{-1} . Although HE has no superiority in rate performance, it significantly improves stability of $\text{Zn}||\text{NaV}_3\text{O}_8$ cells. At a low current density of 0.1 A g^{-1} the specific capacity for the aqueous $\text{Zn}||\text{NaV}_3\text{O}_8$ cell faded highly significantly within 200 cycles, whilst in HE the $\text{Zn}||\text{NaV}_3\text{O}_8$ cell maintained a specific capacity of 164 mAh g^{-1} following 700 cycles (**Supplementary Fig. S22**).’*

Fig. 4. (a) Rate performance for Zn||NaV₃O₈ cells in AE and HE in which the mass loading for NaV₃O₈ is 9.3 mg cm⁻² and the N/P ratio is 5. (b) Energy and power density for Zn||NaV₃O₈ cells based on cathode and anode mass. (c) Cyclic stability and CE for Zn||NaV₃O₈ cells at 0.5 A g⁻¹ in AE and HE.

Supplementary Fig. S22. Cyclic stability and CE for Zn||NaV₃O₈ cells in AE and HE at a current density of 0.1 A g⁻¹.

5. In this context, the measurement in Fig. 4 opens a substantial ambiguity and weakens the seriousness of this paper. Indeed, Fig. 4d and Fig. 4e are not made at the same C-rate (current range). This is understandable because the temperature is not the same, but if the device was

operational at a wide range of temperatures, it should cycle at the same C-rate. Otherwise, it would bias the result to show two different C-rates.

Response

The kinetics for electrode reactions together with ionic conductivity of electrolytes change significantly from 70 to -40 °C. The Zn||NaV₃O₈ cell works at -40 °C when using HE, whilst the cell using AE is not operational at temperature < 0 °C. However, we recognized that HE cannot support a high C-Rate test at a low temperature of -40 °C; therefore, we tested the cell at a low current density of 0.1 A g^{-1} .

We agree with Reviewer #2 that testing at the same C-rate is necessary to avoid bias. To make the comparison reasonable, the Zn||NaV₃O₈ cells in AE or HE were tested at a current density of 0.3 A g^{-1} under varying temperatures. Fig. 5c. shows that HE supported the operation of the Zn||NaV₃O₈ cell in a much wider temperature range from 70 to -40 °C, whilst the aqueous cell is very unstable at 50 and 70 °C, and the lowest operation temperature is 0 °C.

Fig. 5. (c) Specific capacity for NaV₃O₈ electrode at temperature from -40 to 70 °C.

Comment 2-(iv)

Information related to the V_2O_5 cathode. Missing information limits the interest of this electrode.

Response

Thanks for your comment. Information for the cathode is provided, including the XRD pattern, TEM image and XPS spectra.

We must correct a mistake in our MS that the cathode material is not V_2O_5 but in fact NaV_3O_8 . We are misled by two reports, *Nano Energy*, 2016, 22: 583–593 and *Nano Energy*, 2019, 60: 752-759, in which the *as-prepared* material was identified as V_2O_5 . Reviewer #1 pointed out this error, and we therefore conducted additional selected characterizations to confirm that it should in fact, be NaV_3O_8 .

The XRD pattern (**Supplementary Fig. S18c**) identifies the crystalline phase of the *as-synthesized* material as NaV_3O_8 (JCPDS: 16–0601). The elemental mapping images (**Supplementary Fig. S18d**) confirm the existence of Na, V and O in the nanobelt. A 0.77 nm interplanar spacing was observed in the high-resolution TEM image, corresponding to the (001) plane for NaV_3O_8 . The lattice distance agrees well with the 250 °C heated NaV_3O_8 and not the 80 °C heated because the interlayer water was extracted in the high-vacuum sample chamber of TEM.

V 2p XPS spectra evidence the valence variation of vanadium during discharge/charge. In pristine NaV_3O_8 electrode, the majority of vanadium is at 5+ valence and a small portion is at 4+ valence. When discharged at 0.3 V the intensity of the V^{5+} signal decreases whilst that for V^{4+} increases. The peak intensity reversed following charging back to 1.6 V, confirming the electrochemical reversibility of NaV_3O_8 (**Supplementary Fig. S21**).

Supplementary Fig. S18. (a) TEM image for NaV_3O_8 nanobelt. (b) High-resolution TEM image. (c) XRD pattern at 80 °C heated and 250 °C heated NaV_3O_8 . (d) TEM elemental mapping image of NaV_3O_8 nanobelt.

Supplementary Fig. S21. V 2p XPS spectra for NaV₃O₈ electrodes in (a) pristine, (b) discharged at 0.3 V, and (c) charged back to 1.6 V, state.

In response to this comment, we have in our R-MS, Page 12 line 13-16 and line 20-25, included information as follows:

‘The synthesized NaV₃O₈ has a nanobelt morphology (Supplementary Fig. S17) and its interplanar spacing of the (001) plane was measured to be ca. 0.77 nm by high-resolution TEM (Supplementary Fig. S18b). The elemental mapping images (Supplementary Fig. S18d) confirm the uniform distribution of Na, V and O.’

‘The V 2p XPS spectra evidence the valence variation of vanadium during discharge/charge. In pristine NaV₃O₈ electrode, the majority of vanadium is at 5+ valence and a (small) portion part is at 4+ valence. When discharged at 0.3 V the intensity of the V⁵⁺ signal decreases whilst that for V⁴⁺ increases. The peak intensity reversed following charging back to 1.6 V, confirming electrochemical reversibility of NaV₃O₈ (Supplementary Fig. S21).’

1. The current collector for the V₂O₅ electrode is not identified. However, a study of its corrosion (even qualitative) is necessary for the seriousness of the study.

Response

Thank you for your suggestion. The current collector used to support NaV₃O₈ is Ti-foil, which is chemically inert in AE and HE. **Supplementary Fig. S31** shows the morphologies of fresh Ti-foil, and these worked for 300 cycles in AE and HE. There was however no apparent corrosion.

Supplementary Fig. S31. SEM image of (a) fresh Ti-foil and worked for 300 cycles in (b) AE and (c) HE.

2 *The justification of the choice of V_2O_5 does not open the interest of the study. The authors should justify the choice of this electrode material.*

Response

There are several options for cathode materials including, Prussian blue and analogues, organic cathode materials, Mn-based materials and V-based materials. V-based materials usually have greater specific capacity. Although Mn-based materials are practically promising cathodes given relatively high working voltage of 1.2 to 1.4 V, however shortcomings of these materials in aqueous electrolytes cannot be eliminated with hybrid electrolytes. NaV_3O_8 is a practically promising cathode for ZIB because the multivalence of vanadium contributes to high capacity. Importantly, NaV_3O_8 has a large interlayer distance of 0.77 nm, allowing for Zn^{2+} insertion/extraction. Additionally, NaV_3O_8 can be synthesized on a large scale *via* a low-cost, liquid-solid stirring strategy under ambient to providing for large-scale application. Importantly, the NaV_3O_8 electrode works at a high mass loading of $> 9.3 \text{ mg cm}^{-2}$, making it practically possible to increase the energy density of the battery.

The reason for choosing NaV_3O_8 as the cathode is added in the R-MS, Page 12 line 8-12.

' NaV_3O_8 is a promising cathode for ZIB because the multivalence of vanadium contributes to high capacity. More importantly, NaV_3O_8 has a large interlayer distance, 0.77 nm, allowing for Zn^{2+} insertion/extraction. Besides, NaV_3O_8 can be synthesized on a large scale via a low-cost liquid-solid stirring strategy under ambient conditions, providing potential for real applications. Therefore, NaV_3O_8 is selected as the cathode to determine whether HE is compatible in the full cell.'

3. *A 3-electrode study of V_2O_5 , depending on the nature of the electrolyte, would be a necessary added value to understanding the electrochemical dynamics at the cathode. Also, for Zn anode.*

Response

$Zn||NaV_3O_8$ battery is assembled using a 3-electrode battery device (**Supplementary Fig. S20f**) for CV characterizations to determine the electrochemical properties for the NaV_3O_8 electrode.

As is shown in **Supplementary Figs. S20a** and **S20c**, there are two oxidation peaks and three reduction peaks on each curve. It is acknowledged that the relationship between the peak current (i) and scan rates (v) is given by:

$$i = av^b \quad (3)$$

$$\log(i) = b\log(v) + \log(a) \quad (4)$$

where b is the slope of $\log(i)$ vs. $\log(v)$. Both in AE and HE the fitted plots of $\log(i)$ vs. $\log(v)$ (**Supplementary Figs. S20b** and **S20d**) give the b value for the peaks between 0.5 to 1, evidencing that the electrochemical reactions for NaV_3O_8 electrode are composited with ionic diffusion and pseudocapacitance control. To quantify, the capacitive contribution is computed from:

$$i = k_1v + k_2v^{1/2} \quad (5)$$

$$i/v^{1/2} = k_1v^{1/2} + k_2 \quad (6)$$

where the current response of capacitive and ionic diffusion contribution is represented by k_1v and $k_2v^{1/2}$, respectively. **Supplementary Fig. S20e** shows that the percent capacitive contribution increases with increasing scan rate, and the values in AE are overall greater than in HE.

Analysis of the Zn anode is also conducted with the 3-electrode battery device, using well-cleaned Cu foil as working electrode, two Zn foils as counter electrode and reference electrode, respectively. It can be seen in **Supplementary Fig. S5**, the reversibility of Zn plating/stripping reaches 97.1%, higher than that in AE, which is 91.6%.

In response to this comment, we have in our R-MS, Page 12 line 16-20 and Page 7 line 4-5, included discussion as follows:

'The 3-electrode CV curve reveals that there are two oxidation peaks and three reduction peaks during the charging/discharging, and all redox reactions occur between 0.3 V and 1.6 V (Supplementary Fig. S19). Besides, the electrochemical reactions of NaV_3O_8 electrode are composited with ionic diffusion control and pseudo-capacitance control (Supplementary Fig. S20).'

'3-electrode cyclic voltammetry (CV) is conducted to evaluate the reversibility of Zn in AE and CE, respectively. The Coulombic efficiency (CE) of Zn plating/stripping is 97.1% for HE, higher than 91.6% of AE (Supplementary Fig. S5).'

Supplementary Fig. S20. CV characterizations. (a-d) CV curves of NaV₃O₈ electrode at various scan rates and corresponding plots of log (peak current) vs. log (scan rate) in AE and HE, respectively; (e) The comparison of the capacitive contribution at various scan rates in AE and HE, respectively; (f) 3-electrode battery device.

Supplementary Fig. S5. 3-electrode CV curves and the corresponding CE of Zn plating/stripping in (a-b) AE and (c-d) HE. The working electrode is Cu foil, counter and reference electrolyte is Zn foil.

Comment 2-(v)

Minor remarks:

1 The authors write $OTF-1$ instead of $OTF1^-$, in the same way that the Zinc cation is written $Zn2+$ and not $Zn+2$; the authors should correct this minor error.

Response

Thank you for your kind reminder. These errors have been corrected in our R-MS:

- 1) anion is written as OTF^-
- 2) zinc ion is written as Zn^{2+} .

2 Figures 2a, b, c, and d, and figures S12 should be made more prominent as it is difficult to see the SEM images.

Response

In response to this comment, **Figs. 2a, b, c and d**, and **Fig. S12** have been enlarged to make them more prominent. **Fig. S12** has been numbered **Fig. S1**.

Fig. 2. Surface morphology for deposited Zn in (a) AE and (b) HE. (c) Optical microscopy image of Zn/electrolyte interface during Zn deposition in AE and HE. (d) Voltage profile for Zn||Zn symmetric battery tested at, respectively, high current density and high deposition capacity with AE and HE as electrolyte. (g) Surface energy value for main planes of metallic Zn in liquid environment of, respectively, DMAC/TMP/H₂O (5:2:3 by volume), H₂O and DMAC.

Supplementary Fig. S1. Morphology evolution for Zn deposited in 1 M Zn(OTf)₂ solution. Solvent is a DMAC/H₂O mixture with varying percent volume H₂O.

Based on the five family comments, it is difficult to conclude the genuine interest in the present article: performance measures are missing, and information limits the reader's curiosity.

Response

We have responded to and addressed each and all of the comments of Reviewer #2 to provide data on:

1) thermal analysis, 2) conductivity, 3) viscosity and 3) 3-electrode investigation of electrolytes, 4) crystal of the cathode, 5) rate performance, 6) Ragone plot and, 7) temperature adaptability of the Zn||NaV₃O₈ battery, together with details of changes made in our R-MS.

Response to Reviewer #3

Reviewer's Remarks to Authors

In this research, the authors tried to demonstrate the enhanced electrochemical properties of Zn-V₂O₅ battery through controlling the electrolyte composition. However, it seems some critical issues weren't solved to have strong recommendation for publication at Nature Communications.

Response

We thank Reviewer #3 for his/her valuable comments.

Comment 3-1

There is no detailed reason why the authors focused on hybrid electrolyte of dimethylacetamide and trimethyl phosphate. Actually, it was reported usage of dimethylacetamide and trimethyl phosphate can lead to outstanding electrochemical performances of Zn-ion batteries, respectively. (ACS Appl. Mater. Interfaces 2021, 13, 39, 46634–46643, Angew. Chem. Int. Ed., 2019, 58, 2760–2764) Moreover, it seems the related references are not sufficiently cited in this manuscript.

Response

DMAC and TMP were selected as components because they are strong polar molecules. Electron-rich regions of DMAC and TMP, C=O and P=O groups, increase electron density of water protons when interacting with water, and therefore, strengthen the O-H bonds. TMP is an excellent fire retardant that ensures the target electrolyte is non-flammable. Moreover, we found that (002) plane preferred orientation of Zn deposition is achieved if the ratio of DMAC/TMP/H₂O is 5:2:3. We explain that it is because of the modified surface energy of Zn when the hybrid solvent is applied.

DMAC and TMP/TEP have been reported in hybrid electrolytes however, our design principle is different. For *ACS Appl. Mater. Interfaces* 2021, 13, 39, 46634–46643, DMAC is used to increase the number of hydrogen bonds and decrease the number of free O-H bonds, as evidenced by the **redshift** of O-H vibrations in the Raman spectra. In our work, however, the function of DMAC is to increase the O-H bond energy *via* increasing the electron density of water protons. Importantly, the O-H vibrations undergo **blueshift** in HE. The difference between these two functions is in low-temperature performance. In the former design (*ACS Appl. Mater. Interfaces* 2021, 13, 39, 46634–46643), the number of hydrogen bonds will further increase at low temperatures, and therefore the lowest operation temperature of electrolyte is 0 °C. However, HE works well at –40 °C. The different functions derive from salt concentration and the ratio of DMAC/H₂O.

An additional report *Angew. Chem. Int. Ed.*, 2019, 58, 2760-2764, is the first to use TEP fire-retardant as a solvent for Zn anodes, but no detailed discussion on solvation structure was presented however. Importantly, Zn deposition in the electrolyte was a porous structure. However, it is a close-packed morphology with (002) plane preferred crystal orientation in our targeted electrolyte. TMP, together with DMAC alters the surface energy for Zn, and therefore leads to exposure of the (002) plane, although this has yet to be reported.

In response to address this comment of Reviewer #3:

1) The novelty of our MS is:

In reported results, solvents including, DMSO, DMC, DMA, TEP, NMP, EG, PC, DEC and ionic liquid, salts including, high concentration of ZnCl₂, Zn(BF₄)₂, NH₄I and additives including, α -cyclodextrin, glycine, saccharin and La(NO₃)₃ were used to regulate solvation structures, **Supplementary Table S2**. The main mechanisms for these strategies are the breaking of hydrogen between water molecules, restricting the activity of water in solvation sheaths, or excluding water molecules from the electric double layer (EDL), that aim to suppress water decomposition (reduction and oxidation), lower the freezing point of aqueous or aqueous-based electrolytes and boost dendrite-free Zn-plating.

In our work, we demonstrate a new strategy to achieve these goals *via* introducing strong polar molecules, DMAC and TMP, to interact with water and therefore increase electron density of water protons and strengthen O–H bonds of water. These we evidenced *via* DFT, NMR and FT-IR findings. DMAC and TMP alter the surface energy for Zn and therefore guides (002) plane preferred orientation of Zn-deposition. This as a phenomenon has not been reported. Our electrolyte design principle is therefore different it is novel and, is distinguished from reported results.

2) In our R-MS, Page 17 line 8-25, we have included relevant references, a comparison table, and a discussion of findings as follows:

'A comparison of state-of-the-art electrolytes is listed in Supplementary Table S2. For non-aqueous electrolytes, the choices of compatible cathode materials are quite limited and the electrochemical properties, such as specific capacity and lifespan, of these cathodes are inferior compared with aqueous batteries^{17, 22}. As for those aqueous-based electrolytes, in which water accounts for > 50% by volume, the low-rate performance and high-temperature performance of batteries become the shortcomings^{26, 40, 41}, because the interactions between water and non-aqueous solvents are not sufficient to suppress the water activity. In our design, we emphasized the impact of water content on the specific capacity and stability of the NaV₃O₈ cathode and further demonstrated that the stability is highly related to the structural evolution of the cathode during battery operation. Upon comparison, the Zn||NaV₃O₈ battery in HE has advantages at a low cycling current density (0.1 A g⁻¹), and has superior temperature adaptability. By adding non-aqueous solvents to aqueous electrolytes, the improvement in Zn reversibility is quite general, however, the regular morphology of Zn plating is not that universal. Although (002) preferred orientation also occurs in some electrolyte formulas^{23, 26}, but the underlying mechanism has not been well explained. In our work, we confirm that DMAC and TMP could modify the surface energy of Zn and therefore guide the exposure of the (002) plane during Zn plating. DMAC³⁰ and TMP/TEP^{22, 24} have been reported in hybrid electrolytes, however, their functions in strengthening the O-H bonds and guiding the growth of the Zn (002) plane are not reported yet.'

Supplementary Table S2. Comparative summary of state-of-art selected electrolytes for ZIBs.

	Electrolyte	Flammability (Y/N)	Preferred Zn plating orientation	CE for Zn plating-stripping	Oxidation voltage (vs. Zn ²⁺ /Zn)	Cathode	Cycle life	References
1	2 M Zn(OTf) ₂ + 7 M DEC aqueous electrolyte	N	N	Zn Cu cell, 99.24 %, 400 cycles, 1 mA cm ⁻² , 1 mAh cm ⁻²	N/A	V ₂ O ₅ ·nH ₂ O (mass loading N/A)	202 mAh g ⁻¹ with 72.9 % retention (2 A g ⁻¹), 5000 cycles (RT)	ACS Nano , 2022, 16, 9667–9678.
2	0.5 M Zn(OTf) ₂ in TMP/DMC	N	N	Zn SS cell, 99.15 %, 300 cycles, current and capacity N/A	2.25 V	VS ₂ , 4 mg cm ⁻²	107.86 mAh g ⁻¹ (100 mA g ⁻¹), 500 cycles (RT)	Adv. Mater. , 2019, 31, 1900668.
3	3 M Zn(CF ₃ SO ₃) ₂ in PC/H ₂ O (2:8)	N	N	N/A	N/A	NaV ₃ O ₈ ·1.5 H ₂ O, 8 mg cm ⁻²	168 mAh g ⁻¹ (0.2 A g ⁻¹), 400 cycles, -40 °C 229 mAh g ⁻¹ (5 A g ⁻¹), 1000 cycles, 30 °C	Adv. Funct. Mater. , 2022, 32, 2111714.
4	0.5 M Zn(CF ₃ SO ₃) ₂ in TEP/H ₂ O (7:3)	N	N	Zn SS cell, 93.71 %, 100 cycles, current and capacity N/A	2.25 V	KCuHCF, 1 mg cm ⁻²	~50 mAh g ⁻¹ , 74 % retention, 1000 cycles, average CE 97.66 %	Angew. Chem. Int. Ed. , 2019, 58, 2760–2764.
5	4 M Zn(TFSI) ₂ + 4 M P ₄₄₄ (201)-TFSI in H ₂ O	N/A	N	Zn Cu cell, ~99 %, 1.17 mA cm ⁻² , 1.17 mAh cm ⁻² , 20% Zn utilization, 16 cycles	2.42 V	Na ₂ V ₆ O ₁₆ ·1.6 3H ₂ O, 1 mg cm ⁻²	~120 mAh g ⁻¹ , 1900 cycles (300 mA g ⁻¹) (RT)	Angew. Chem. Int. Ed. , 2021, 60, 12438–12445.
6	1 M Zn(CF ₃ SO ₃) ₂ in PC/H ₂ O (5:5)	N	Y	Zn Cu cell, 99.93 %, 1 mA cm ⁻² , 0.5 mAh cm ⁻² , 500 cycles	~ 2.5 V	PANI, 1.5-2 mg cm ⁻²	78 mAh g ⁻¹ (0.2 A g ⁻¹), 200 cycles, -20 °C 95 mAh g ⁻¹ (0.2 A g ⁻¹), 200 cycles, 50 °C	J. Am. Chem. Soc. , 2022, 144, 7160–7170.
7	3 M ZnSO ₄ + 10 mM α-cyclodextrin aqueous electrolyte	N	Y	Zn Cu cell, ~99.9 %, 1 mA cm ⁻² , 1 mAh cm ⁻² , 600 cycles	~ 2.5 V	V ₂ O ₅ , 6.4 mg cm ⁻²	~180 mAh g ⁻¹ , 200 cycles (1 A g ⁻¹) (RT)	J. Am. Chem. Soc. , 2022, 144, 11129–11137.
8	2 M ZnSO ₄ and 0.0085 M La(NO ₃) ₃ aqueous electrolyte	N	Y	Zn Ti cell, ~99.9 %, 2 mA cm ⁻² , 1 mAh cm ⁻² , 2200 cycles	N/A	VS ₂ , 8 mg cm ⁻² (N/P ratio 4:3)	120 mAh g ⁻¹ (100 mA g ⁻¹), 100 cycles (RT)	Nat. Commun. , 2022, 13, 3252.

9	1 mol/kg Zn(TFSI) ₂ + 20 mol/kg LiTFSI aqueous electrolyte	N	N	Zn Pt (three electrode cell), 99.7 %	N/A	LiMn ₂ O ₄ , 2.4 mAh cm ⁻²	26 mAh g ⁻¹ (4 C), 4000 cycles (RT)	Nat. Mater. , 2018, 17, 543–549.
10	2 M ZnSO ₄ in NMP/H ₂ O (5:5)	N/A	Y	Zn Cu cell, ~99.7 %, 1 mA cm ⁻² , 0.5 mAh cm ⁻² , 1000 cycles	N/A	VS ₂ , 5.3 mg cm ⁻²	125 mAh g ⁻¹ (1 A g ⁻¹), 2000 cycles (RT)	Adv. Energy Mater. , 2022, 12, 2103231.
11	3 M ZnSO ₄ + 0.5 M glycine aqueous electrolyte	N	Y	Zn Cu cell, 99.68 %, 2 mA cm ⁻² , 2 mAh cm ⁻² , 650 cycles	N/A	NH ₄ V ₄ O ₁₀ (mass loading N/A)	220 mAh g ⁻¹ (5 A g ⁻¹), 3000 cycles (RT)	ACS Nano , 2022, DOI: 10.1021/acsnano.2c09317
12	1 M ZnAc ₂ + 4 M NH ₄ I aqueous electrolyte	N	N	Zn Cu cell, 99.8 %, 1 mA cm ⁻² , 1 mAh cm ⁻² , 100 cycles	N/A	I ₂ (mass loading N/A)	~1 mAh cm ⁻² , 200 cycles	J. Am. Chem. Soc. , 2022, 144, 18435–18443
13	1.3 M ZnCl ₂ in H ₂ O/DMSO (volume ratio of H ₂ O/DMSO = 4.3:1)	N/A	N	Zn Cu cell, 99.5 %, 1 mA cm ⁻² , 0.5 mAh cm ⁻² , 400 cycles	~2.2 V	MnO ₂ (mass loading N/A)	150 mAh g ⁻¹ (8 C), 500 cycles (RT)	J. Am. Chem. Soc. , 2020, 142, 21404–21409.
14	BMITFSI:Zn(TFSI) ₂ (water/ionic liquid mass 20 %)	N	N	Zn Cu cell, 99.27 %, 1 mA cm ⁻² , 0.5 mAh cm ⁻² , 400 cycles	~2.5 V	PANI, 0.8-0.95 mg cm ⁻²	18 mAh g ⁻¹ (1 A g ⁻¹), 1000 cycles (RT)	ACS Energy Lett. , 2023, 8, 608–618.
15	2 M ZnSO ₄ in water/EG (volume ratio of EG 40 %)	N	N	Zn Ti cell, ~98 %, 2 mA cm ⁻² , 1 mAh cm ⁻² , 120 cycles	~2.8 V	PANI/V ₂ O ₅ , 2.5-3.0 mg cm ⁻²	~45 mAh g ⁻¹ (0.2 A g ⁻¹), 6500 cycles, -20 °C ~60 mAh g ⁻¹ (5 A g ⁻¹), 50000 cycles, 20 °C	Energy Environ. Sci. , 2020, 13, 3527–3535.
16	1 M ZnTFMS/DMF	Y	N	Zn SS cell, ~99.8 %, 1 mA cm ⁻² , 1 mAh cm ⁻² , 200 cycles	~2.4 V	PQMCT (mass loading N/A)	~22 mAh g ⁻¹ (0.2 A g ⁻¹), 1 cycle, -70 °C ~180 mAh g ⁻¹ (2 A g ⁻¹), 1 cycle, 150 °C	Angew. Chem. Int. Ed. , 2020, 59, 14577 – 14583
17	7.5 mol/Kg ZnCl ₂ -based aqueous electrolyte	N/A	N	Zn Cu cell, ~99 %, 0.2 mA cm ⁻² , 0.2 mAh cm ⁻² , 50 cycles	~1.9 V	PANI, (mass loading N/A)	~75 mAh g ⁻¹ (0.2 A g ⁻¹), 2000 cycles, -70 °C	Nat Commun. , 2022, 11, 4463.
18	4 M Zn(BF ₄) ₂ aqueous electrolyte	N	N	Zn SS cell, ~95 %, 0.5 mA cm ⁻² , 0.5 mAh, 1 cycle	~2 V	TCBQ2, 1-2 mg	~95 mAh g ⁻¹ (1 C), 1000 cycles, -30 °C ~80 mAh g ⁻¹ (0.1 C), 50 cycles, -30 °C	J. Mater. Chem. A , 2021, 9, 7042–7047.

19	2 M Zn(OTf) ₂ in water/DMC (volume ratio 4:1)	N	N	Zn Ti cell, ~99.8 %, 1 mA cm ⁻² , 1 mAh cm ⁻² , 200 cycles	N/A	V ₂ O ₅ , 2 mg cm ⁻²	~380 mAh g ⁻¹ (2 A g ⁻¹) 1000 cycles, RT	Chem. Sci. , 2021, 12, 5843–5852.
20	2.5 M Zn(NO ₃) ₂ +13 M LiNO ₃ in DMAC/H ₂ O	N/A	N	N/A	~1.9 V	LiMn ₂ O ₄ , 3 mg cm ⁻²	95 mAh g ⁻¹ (1 C), 200 cycles (RT)	ACS Appl. Mater. Interfaces , 2021, 13, 39, 46634–46643.
21	1 M Zn(OTf) ₂ in DMAC/TMP/H ₂ O (volume ratio 5:2:3)	N	Y	Zn Cu cell, 99.5 %, 1 mA cm ⁻² , 1 mAh cm ⁻² , 2000 cycles	~2.25 V	NaV ₃ O ₈ ~6 mg cm ⁻² for RT testing; ~0.6 mg cm ⁻² for high-low temperature testing	~264 mAh g ⁻¹ (0.1 A g ⁻¹) 700 cycles, (RT); ~190 mAh g ⁻¹ (0.5 A g ⁻¹) 3000 cycles, (RT); ~100 mAh g ⁻¹ (0.1 A g ⁻¹) 1800 cycles, -40 °C; ~200 mAh g ⁻¹ (5 A g ⁻¹) 1400 cycles (retention ~60 %), 70 °C;	This work

Comment 3-2

It is also required to perform the electrochemical tests under other DMAC/TMP/H₂O electrolytes with different volume ratio. Although each case of TMP/H₂O and DMAC/H₂O were studied on an effect of different volume ratio, it is insufficient to explain the reason on usage of DMAC/TMP/H₂O (5:2:3 by volume). The authors should show why the volume ratio of 5:2:3 (=DMAC/TMP/H₂O) was used as the optimal electrolyte composition for this research.

Response

5:2:3 is the optimized ratio for DMAC/TMP/H₂O solvent based on electrochemical performance of the electrodes and safety.

As is shown in **Supplementary Fig. S1**, the (002) preferred orientation of Zn deposition is achieved when the volume ratio of H₂O in the DMAC/H₂O mixture is in the range of 20 to 40%. The ratio of H₂O has significant impact on the specific capacity and stability of the NaV₃O₈ cathode. With 20 % water the specific capacity for NaV₃O₈ is (relatively) low, and with 40 % specific capacity is greater in the first few cycles, but the rate of decay is faster, **Supplementary Fig. S2**. Therefore, 30 % water is the 'best' option.

However, the DMAC/H₂O mixture is flammable. Fire retardant TMP is used to replace partly DMAC to ensure non-flammability of the electrolyte however, the amount needs to be minimized to maintain low viscosity of the electrolyte. The data of **Supplementary Fig. S3** confirms that minimum usage of TMP is around 30 % in the mixture of DMAC/TMP. Therefore the ratio of TMP in DMTC/TMP/H₂O is 70 % × 30 % = 21 %. Considering Zn deposition morphology, capacity of electrode and non-flammability of electrolyte, we optimized electrolyte with DMAC/TMP/H₂O of 5 (~70 % × 70 %):2 (~70 % × 30 %): 3 (30 %).

Significantly, this optimized electrolyte cannot be ignited, the Coulombic efficiency (CE) for Zn plating/stripping reaches an average value of CE = 99.5 % together with a dendrite-free Zn deposition. The specific capacity of cathode is *ca.* 200 mAh g⁻¹ at a charge/discharge current density of 0.5 A g⁻¹, and the cycling life of the full cell is significantly extended to 3000 cycles.

Supplementary Fig. S1. Morphology evolution for Zn deposited in 1 M $\text{Zn}(\text{OTf})_2$ solution. Solvent is a DMAC/ H_2O mixture with varying percent volume H_2O .

Supplementary Fig. S2. Cyclical performance for $\text{Zn}||\text{NaV}_3\text{O}_8$ cells in 1 M $\text{Zn}(\text{OTf})_2$ solution. Solvent is DMAC/ H_2O mixture with varying percent volume H_2O .

Supplementary Fig. S3. Digital images of ignition test for 1 M $\text{Zn}(\text{OTf})_2$ solution. The Solvent is (a) DMAC, (b) DMAC/TMP (8:2 by volume), (c) DMAC/TMP (7:3 by volume), (d) DMAC/ H_2O (5:3 by volume), (e) DMAC/TMP/ H_2O (5:2:3 by volume).

In response to this comment, we have in our R-MS, Page 5 line 18-27, included the explanatory text, namely:

‘Electrolyte formulation was based on three (3) principles, namely: 1) dendrite-free deposition of Zn; 2) good compatibility with NaV_3O_8 cathode; and 3) non-flammability. It was observed that the Zn deposition orientates the (002) plane when the volume percentage of water in DMAC/ H_2O mixture ranged from 20 to 40 % (Supplementary Fig. S1). Water content has a significant impact on cathode performance, and the NaV_3O_8 electrode exhibited a satisfactory capacity and cyclic stability when the ratio of H_2O was 30 % (Supplementary Fig. S2). TMP was added to ensure the formulated electrolyte is nonflammable, with its minimum use ca. 30% in the mixture of DMAC/TMP (Supplementary Fig. S3). Therefore, the volume ratio of DMAC/TMP/ H_2O was determined to be 5:2:3. 1 M $\text{Zn}(\text{OTf})_2$ dissolved in this mixture is denoted HE whilst 1 M $\text{Zn}(\text{OTf})_2$ aqueous solution, AE.’

Comment 3-3

Actually, the research on Zn-V2O5 battery was already reported in many papers. Thus, it is also very important to confirm the target material can exhibit the general electrochemical performances of Zn-V2O5 battery. However, the author didn't present the charge/discharge curves and CV curves of Zn-V2O5 battery measured at a low current density with room temperature. Moreover, there is no data showing the redox reaction of vanadium ions during charge and discharge.

Response

Thank you for your comments. We must correct a mistake in our MS that the cathode material is not V_2O_5 but in fact NaV_3O_8 . We were misled by two reports, *Nano Energy*, 2016, 22: 583–593 and *Nano Energy*, 2019, 60: 752-759, in which the *as-prepared* material was identified as V_2O_5 . Reviewer #1 pointed out this error, and we therefore conducted additional selected characterizations to confirm that it should in fact, be NaV_3O_8 .

Supplementary Fig. S23 presents the charge/discharge curves for $Zn||NaV_3O_8$ cell at room temperature. In AE, the curves vibrate and the voltage decays significantly following 300 cycles, evidencing the coming failure of the battery. However, this did not occur in HE even following 3000 cycles. CV was conducted to determine the electrochemical properties of the NaV_3O_8 electrode. **Fig. 4b** shows that there are two oxidation peaks and three reduction peaks, and that they are reversible. Long-term cycling of full cells tested at low current density $0.1 A g^{-1}$ have been added in **Supplementary Fig. S22**. The specific capacity of the aqueous $Zn||NaV_3O_8$ cell faded significantly within 200 cycles, whilst in HE it continued to exhibit a specific capacity of $164 mAh g^{-1}$ following 700 cycles, confirming that the NaV_3O_8 cathode is more stable in HE. The V 2p XPS spectra confirm the valence variation for vanadium during discharge/charge. In pristine NaV_3O_8 electrode, the majority of vanadium is at 5+ valence and a small portion at 4+ valence. When discharged at 0.3 V the intensity of the V^{5+} signal declines while that for V^{4+} increases. The peak intensity reversed following charging back to 1.6 V, confirming the electrochemical reversibility of NaV_3O_8 (**Supplementary Fig. S21**).

In response to this comment, we have therefore included additional information in our R-MS, Page 13 line 6-10, Page 12 line 16-20, Page 13 3-6, and Page 12 line 20-25, as follows:

*‘At a current density of $0.5 A g^{-1}$, in AE, the NaV_3O_8 cathode exhibits a rapid increase in specific capacity in the first 20 cycles from 170 to 231 $mAh g^{-1}$ and, an apparent decline in subsequent cycling to fail completely in the following 400 cycles. Corresponding charge-discharge curves evidence that the voltage decays highly significantly following 300 cycles (**Supplementary Fig. S23**).’*

*‘The 3-electrode cyclic voltammetry (CV) curve reveals that there are two oxidation peaks and three reduction peaks during the charge/discharge, and all redox reactions occur between 0.3 and 1.6 V (**Supplementary Fig. S19**). Additionally, the electrochemical reactions for NaV_3O_8 electrode are composited with ionic diffusion control and pseudo-capacitance control (**Supplementary Fig. S20**).’*

*‘At a low current density $0.1 A g^{-1}$ the specific capacity for aqueous $Zn||NaV_3O_8$ cell faded significantly within 200 cycles, whilst in HE, it continued to exhibit a specific capacity of $164 mAh g^{-1}$ following 700 cycles (**Supplementary Fig. S22**).’*

*‘The V 2p XPS spectra confirm the valence variation for vanadium during discharge/charge. In pristine NaV_3O_8 electrode the majority of vanadium is at 5+ valence and a (small) portion at 4+ valence. When discharged at 0.3 V the intensity of the V^{5+} signal declines whilst that for V^{4+} increases. The peak intensity reversed following charging back to 1.6 V, confirming the electrochemical reversibility of NaV_3O_8 (**Supplementary Fig. S21**).’*

Supplementary Fig. S23. Charge-discharge curves for Zn||NaV₃O₈ battery in (a) AE and (b) HE.

Fig. 4. (b) 3-electrode CV curves for NaV₃O₈ electrode in HE;

Supplementary Fig. S22. Cyclic stability and CE for Zn||NaV₃O₈ cells in AE and HE at a current density of 0.1 A g⁻¹.

Supplementary Fig. S21. V 2p XPS spectra for NaV₃O₈ electrodes in (a) pristine, (b) discharged at 0.3 V, and (c) charged back to 1.6 V, state.

Comment 3-4

Several SEM results show the electrolyte affected the morphology of Zn-foil after cycling. These results imply the main effect of the electrolyte for electrochemical performances of Zn-V₂O₅ battery

may be attributed to the morphology of Zn-foil, not structural evolution of V₂O₅. The authors should explain what is the main effect to determine the enhanced electrochemical performances of Zn-V₂O₅ battery between the morphology of Zn-foil and structural evolution of V₂O₅ in detail.

Response

The failure of Zn||NaV₃O₈ cell is attributed to (at least) one of the following: 1) short-circuit caused by Zn dendrite growth, 2) exhaustion of Zn anode, or electrolyte, 3) poor electrical contact caused by gas generation induced battery deformation, and/or 4) cathode degradation.

For the cells with AE, at a low rate of 0.1 A g⁻¹ dendritic growth slows and the Zn anode/electrolyte consumes at a low rate. Therefore, cathode degradation becomes the primary reason for failure of Zn||NaV₃O₈ cell. As is shown in **Supplementary Fig. S24a**, decreased specific capacity of the aqueous full cell cannot be recovered by replacing the Zn anode and refilling the electrolyte. This changes when the full cell works at a medium current rate of 0.5 A g⁻¹. Dendrite growth becomes the ‘trigger’ to invalidate the Zn||NaV₃O₈ cell. It can be seen from the disassembled battery, **Supplementary Fig. S24c**, that the dendrites penetrated the separator following 200 cycles. The cathode electrode was taken out and reassembled with fresh Zn anode, and a new separator and sufficient electrolyte, importantly however, specific capacity failed to recover to its initial value (**Supplementary Fig. S24b**). Cathode decay was still observed even though its decay rate was less than that at a low current rate.

With HE dendrite growth is addressed because of (002) preferred orientation of Zn deposition. The high Coulombic efficiency (CE) for Zn plating/stripping minimizes consumption of both Zn anode and electrolyte. Moreover, a better structural stability for NaV₃O₈ electrode contributes to good capacity retention of the full cell. HE exhibits good overall performance, both on the cathode and Zn anode, accounting for the superiority of the full cell performance compared with AE.

Supplementary Fig. S24. Failure analysis of Zn||NaV₃O₈ batteries tested in AE. (a) The Battery tested at 0.1 A g⁻¹ and Zn anode replaced and electrolyte refilled at 182nd cycle. (b) Battery tested at 0.5 A g⁻¹ and Zn anode replaced and electrolyte refilled at 207th cycle. (c) Battery disassembled following short-circuit. (d-e) Reference battery tested in HE.

In response therefore to address fully this comment of Reviewer #3 to explain the main effect to determine enhanced electrochemical performance we have included in our R-MS, Page 13 line 11-16, the following explanatory text, namely:

‘To determine the reason for aqueous Zn||NaV₃O₈ battery failure, NaV₃O₈ electrodes were extracted when capacity decreased significantly and reassembled with fresh Zn anode and sufficient electrolyte. It was found that following the refresh specific capacity for NaV₃O₈ electrodes could not be recovered to that of its initial state, evidencing that cathode degradation significantly affects battery performance in AE (Supplementary Fig. S24).’

END OF RESPONSE TO REVIEWS

REVIEWER COMMENTS

Reviewer #1 (Remarks to the Author):

The authors made great efforts to revise this work, and they provided reasonable explanations and responses to the reviewers' comments. All my original concerns are now well addressed. Thus, I strongly recommend this work for publication.

Reviewer #2 (Remarks to the Author):

The authors have made several changes to this new and improved text version, including some previously missing but crucial components. To summarize my main complaint points as Reviewer 2, they are as follows: (iii) metrics for success, and (iv) background on V2O5. Therefore, I appreciate the authors' efforts to address these two comments, which are crucial to the study's validity. Regarding major criticism (iv), the authors have greatly improved the version.

(a) Regarding primary objection (iii), Figure 4 B must be compared to the current state of the art. Previous research will help to determine if the present outcome is satisfactory.

(b) Unfortunately, the critical corrosion issue was only superficially addressed. As the authors introduce it, it is a significant problem of this technology. The study of corrosion in the new electrolyte must be addressed in depth.

(c) Reading the response to the reviewers, I agree with the comments of reviewer three regarding the novelty of the materials tested. The authors' response is weak on this point, and it is a crucial weakness concerning the impact of this scientific article.

(d) Another remark concerns the quality of the figures. This is a formal weakness that the authors should improve because the article loses readability, especially for figure 5, which is too full of information and is in small type.

In conclusion, weaknesses b and c are too strong to accept the article, despite the technical quality of the results. The novelty of the work is an indispensable criterion for validation in such a journal as Nature Communication.

Reviewer #3 (Remarks to the Author):

The revised manuscript satisfies all comments of the reviewer, thus, I recommend the acceptance of this manuscript for publication

REVIEWER COMMENTS

Reviewer #1 Remarks to Author

The authors made great efforts to revise this work, and they provided reasonable explanations and responses to the reviewers' comments. All my original concerns are now well addressed. Thus, I strongly recommend this work for publication.

Response

We thank Reviewer #1 for his/her recommendation.

Reviewer #2 Remarks to Author

The authors have made several changes to this new and improved text version, including some previously missing but crucial components. To summarize my main complaint points as Reviewer 2, they are as follows: (iii) metrics for success, and (iv) background on NaV_3O_8 . Therefore, I appreciate the authors' efforts to address these two comments, which are crucial to the study's validity. Regarding major criticism (iv), the authors have greatly improved the version.

Comment 2-1

(a) Regarding primary objection (iii), Figure 4 B must be compared to the current state of the art. Previous research will help to determine if the present outcome is satisfactory.

Response

We agree with Reviewer #2.

In response to address fully this comment, we have compared our results with those in which sodium vanadates are used as cathode materials. Because many reports however did not explicitly give energy density and power density, we estimated values based on given charge/discharge curves, current density and specific capacity.

Findings are summarized in **Table S1**. These data we added in a Ragone plot to give a direct comparison. As shown in **Fig. S24c**, both the $\text{Zn}||\text{NaV}_3\text{O}_8$ batteries using AE and HE as electrolyte exhibit comparable energy density. At low current density, the batteries with AE and HE exhibit similar energy density, however, the battery with AE exhibits superiority at high current density. The lifespan of reported battery systems is poor at low and medium rates, and available data for comparison is limited. It can be seen in **Fig. S29** that HE exhibits significant advantages in supporting operation of $\text{Zn}||\text{NaV}_3\text{O}_8$ at low and medium current density.

Supplementary Fig. S24. Charge-discharge curves for Zn||NaV₃O₈ battery in (a) AE and (b) HE tested at current density, respectively, 0.1, 0.2, 0.5, 1, 2 and 5 A g⁻¹. (c) Comparison of energy density/power density of this work with reported findings based on mass of cathode active materials.

Supplementary Fig. S29. Comparison of battery lifespan with reported results based on data of Table S1.

Because most of the reported data were obtained from batteries with low cathode material loading, flood electrolyte and thick Zn anode, the energy density is calculated based on the mass of cathode active material only. An excess Zn-foil is used in the test in aqueous batteries because of corrosion of the Zn anode, and the mass of the Zn anode is usually excluded from

the calculation for energy density. In HE Zn corrosion is significantly suppressed, therefore the areal capacity ratio of negative to positive electrodes (N/P ratio) can be controlled at 5. Additionally, a thick NaV_3O_8 electrode with a mass loading of 9.2 mg cm^{-2} was used as in the cathode. The capacity of the thick NaV_3O_8 electrode is *ca.* 1.9 mAh (6.5 mg of NaV_3O_8) and that for Zn anode is *ca.* 9.5 mAh (11.6 mg). **Fig. 4a** presents the rate performance for the thick NaV_3O_8 electrode in AE and HE. Based on the mass of the cathode and Zn anode, the energy density for $\text{Zn}||\text{NaV}_3\text{O}_8$ batteries in AE can reach 86 Wh kg^{-1} , whilst that in HE is lower at 77 Wh kg^{-1} . This value is meaningfully greater than that for the reported $\text{Zn}||\text{NaV}_3\text{O}_8$ battery, which uses 1 M ZnSO_4 aqueous solution as electrolyte and $1 \text{ M Na}_2\text{SO}_4$ as additive [*Nat Commun.*, 2018, 9, 1656].

Because of the lack of energy density data based on the mass of cathode and anode in reports for Zn-ion batteries, we compared our findings with those in aqueous Li-ion and Na-ion batteries, as is shown in **Fig. S25**. It can be seen that the energy density for our $\text{Zn}||\text{NaV}_3\text{O}_8$ battery in HE surpasses that for the majority of aqueous Li-ion and Na-ion batteries, and approaches the value of the *state-of-the-art* aqueous Li-ion batteries. $\text{Zn}||\text{NaV}_3\text{O}_8$ battery is apparently more competitive than aqueous Li-ion batteries when affordability is taken into account.

Fig. 4. (a) Rate performance for $\text{Zn}||\text{NaV}_3\text{O}_8$ cells. (b) Energy and power density for $\text{Zn}||\text{NaV}_3\text{O}_8$ cells based on the total mass of cathode and anode electrode.

Supplementary Fig. S25. Comparison of energy density for $\text{Zn}||\text{NaV}_3\text{O}_8$ battery in HE with selected aqueous batteries based on the mass of cathode and anode active materials. Detailed data are in **Table S2**.

Supplementary Table S1. Comparison of energy density and power density for ZIBs based on mass of cathode active materials.

No.	Cathode material	Electrolyte	Mass loading	Energy density (Wh kg ⁻¹)	Power density (W kg ⁻¹)	Current density (A g ⁻¹)	Lifespan (cycle)	Reference
1	NaV ₃ O ₈	1 M ZnSO ₄ /1 M Na ₂ SO ₄ aqueous solution	1.2 mg cm ⁻²	307	75	5	2000	Science China Chemistry , 2019, 62, 609–615
				262	150			
				225	375			
				187	720			
				151	1440			
				126	2100			
				112	2800			
				105	3500			
2	NaV ₃ O ₈	1 M ZnSO ₄ /1 M Na ₂ SO ₄ aqueous solution	2 mg cm ⁻²	270	71	1	100	Nat. Commun. , 2018, 9, 1656
				227	142	4	1000	
				196	350			
				168	700			
				150	1500			
				128	3000			
3	NaV ₃ O ₈	3 M ZnSO ₄ aqueous solution	2.4 mg cm ⁻²	212	40	0.05	300	ACS Appl. Mater. Interfaces 2020, 12, 54627–54636
				168	80	0.4	2000	
				143	168			
				110	324			
				77.5	608			
				66	730			
4	Na ₂ V ₆ O ₁₆ · 3H ₂ O	1.0 M Zn(ClO ₄) ₂ in propylene carbonate	N/A	231	340	0.5	100	Batteries & Supercaps 2020, 3, 254-260
				224	560	2	5000	
				204	660	5	5000	

				161	1240			
				125	1710			
				78	2350			
				188	77	4	2000	
				167	158			
5	$\text{Na}_2\text{V}_6\text{O}_{16} \cdot n\text{H}_2\text{O}$	0.5m $\text{Zn}(\text{ClO}_4)_2$ with 18m NaClO_4 aqueous solution	3~4 mg cm^{-2}	130	390			Energy Environ. Sci. , 2021, 14, 4463
				108	800			
				88	1600			
				74	3200			
				317	72	0.1	45	
6	$\text{Na}_5\text{V}_{12}\text{O}_{32} \cdot 11.9\text{H}_2\text{O}$	3 M $\text{Zn}(\text{CF}_3\text{SO}_3)_2$ aqueous solution	1.1~3.4 mg cm^{-2}	274	144	1	3800	Mater. Today Energy , 2021, 21, 100757
				244	370			
				190	760			
				148	1480			
				220	228	0.1	50	
7	$\text{Na}_5\text{V}_{12}\text{O}_{32}$	3 M ZnSO_4 aqueous solution	~2 mg cm^{-2}	185	380	0.5	100	Adv. Energy Mater. , 2018, 8, 1801819
				151	760	4	2000	
				92	1520			
				45.6	3800			
				266	76	0.1	50	
8	Mn-Doped $\text{Na}_5\text{V}_{12}\text{O}_{32}$	2 M $\text{ZnSO}_4/0.5$ M Na_2SO_4 aqueous solution	~1.5 mg cm^{-2}	243	152	1	250	Energy Fuels , 2021, 35, 13483–13490
				228	380			
				176	760			
				162	1520			
				82	3800			

9	NaV ₃ O ₈	AE	~1.8 mg cm ⁻²	248	70	0.1	100	This work
				235	154	0.5	400	
				205	400	4	6000	
				182	800			
				159	1600			
				122	3800			
10	NaV ₃ O ₈	HE	~1.8 mg cm ⁻²	260	70	0.1	700	This work
				244	152	0.5	3000	
				204	380	4	6000	
				170	750			
				129	1400			
				74	2800			

Supplementary Table S2. Comparison of energy density with reported aqueous Li-ion and Na-ion batteries.

No	Cathode/anode electrodes	Cathode/anode mass ratio	Average voltage	Energy density based on mass of cathode and anode (Wh kg ⁻¹)	Reference
1	LiTi ₂ (PO ₄) ₃ /C LiMn ₂ O ₄	1.5:1	1.55 V	68	Sci Rep , 2015, 5, 10733
2	TiS ₂ LiMn ₂ O ₄	2:1	N/A	78	Electrochem. Commun. , 2017, 82, 71-74
3	VO ₂ LiMn ₂ O ₄	N/A	1.5 V	55	Science , 1994, 264,1115-1118
4	LiV ₃ O ₈ LiNi _{0.81} Co _{0.19} O ₂	N/A	1~1.2 V	54	Electrochimica Acta , 2000, 46, 59–65
5	Mo ₆ S ₈ LiMn ₂ O ₄	2:1	1.5 V and 2 V	84	Science , 2015, 350, 938-943
6	acetylene black LiMn ₂ O ₄	1:2	~1.3 V	35	J. Electrochem. Soc. ,2006, 153, A450–A454
7	LiTi ₂ (PO ₄) ₃ LiFeO ₄	1:1	0.9 V	50	Nature Chemistry , 2010, 2, 760–765
8	LiTi ₂ (PO ₄) ₃ LiMn ₂ O ₄	1:1	1.5 V	60	Adv. Funct. Mater. , 2007, 17, 3877–3884
9	Polyimide LiCoO ₂	1.2:1	1.12 V	80	J. Power Sources , 2014,249,367–372
10	Na ₃ MnTi(PO ₄) ₃ Na ₃ MnTi(PO ₄) ₃	1:1	1.4 V	40	Angew. Chem. Int. Ed. , 2016, 55, 12768–12772
11	polymerized pyrene-4,5,9,10-tetraone Na ₃ V ₂ (PO ₄) ₃	N/A	0.65 V	30	Nat. Mater. , 2017, 16, 841–848
12	Polyimide NaI	N/A	0.8 V	63.8	Sci. Adv. , 2016, 2, e1501038
13	NaTi ₂ (PO ₄) ₃ Na _{0.44} MnO ₂	2.5:1	1.1 V	33	Adv. Energy Mater. , 2013, 3,290–294
14	Zn NaV ₃ O ₈	N/A	0.75	70	Nat Commun. , 2018, 9, 1656
15	Zn NaV ₃ O ₈	1:1.8	0.75 V	77	This work

We have in our R-MS, Page 12 line 20-35 and Page 13 line 1-11, included the comparison and discussion as follow:

At low current density, the batteries with AE and HE show similar energy density, however, the battery with AE exhibits superiority at high current densities, (Supplementary Fig. S24). To evaluate the energy density at the electrode scale, Zn||NaV₃O₈ cells are assembled with a thick cathode and a thin Zn foil, in which the cathode mass loading is 9.3 mg cm⁻² and the areal capacity ratio of negative to positive electrodes (N/P ratio) is 5. Because of high ionic conductivity of AE, the Zn||NaV₃O₈ cell exhibits better rate performance, with ca. 80 mAh g⁻¹ at 5A g⁻¹, as is seen in Fig. 4a. The Zn||NaV₃O₈ cell using HE does not exhibit an advantage in specific capacity, especially when the applied current density increases. The energy density and power density of Zn||NaV₃O₈ batteries are given according to the rate performance, calculated with the mass of the cathode and anode (Fig. 4b). The battery in AE exhibits a superior energy density of 86.1 Wh kg⁻¹ and greater power energy density of 1019 W kg⁻¹, while in HE, the energy density is 77 Wh kg⁻¹ and the power maximum is 665 W kg⁻¹. The energy density of Zn||NaV₃O₈ battery in HE surpasses that of the majority of aqueous Li-ion and Na-ion batteries, and approaches the value of the state-of-the-art Li-ion batteries (Supplementary Table S2 and Fig. S25). However, the Zn|| NaV₃O₈ battery appears to be far more competitive than aqueous Li-ion batteries if affordability is taken into consideration. Although HE has no superiority in rate performance, it significantly improves stability of Zn||NaV₃O₈ cells. At a low current density of 0.1 A g⁻¹ the specific capacity for the aqueous Zn||NaV₃O₈ cell faded significantly within 200 cycles, whilst in HE the Zn||NaV₃O₈ cell maintained a specific capacity of 164 mAh g⁻¹ following 700 cycles (Supplementary Fig. S26). At a current density of 0.5 A g⁻¹, in AE, the NaV₃O₈ cathode exhibits a rapid increase in specific capacity in the first 20 cycles from 170 to 231 mAh g⁻¹, and an apparent decline in subsequent cycling, to fail completely in the following 400 cycles, Fig. 4c. The corresponding charge-discharge curves show that the voltage decay remarkably after 300 cycles (Supplementary Fig. S27). In contrast, the specific capacity for the NaV₃O₈ cathode in HE remains stable and has a lifespan of 3000 cycles.”

Comment 2-2

(b) Unfortunately, the critical corrosion issue was only superficially addressed. As the authors introduce it, it is a significant problem of this technology. The study of corrosion in the new electrolyte must be addressed in depth.

Response

We agree with Reviewer #2 that corrosion in the new electrolyte must be addressed.

In response, a requirement for an adequate electrolyte is that it be compatible with all components of the battery, especially the current collector and Zn anode.

Because during operation of ZIBs, the applied voltage on Ti-foil can be up to 1.8 V, the corrosion of the Ti current collector was therefore determined *via* measuring leakage current during voltage hold. Fig. S35a-c evidences that the leakage current remains low for both AE and HE at a high voltage of 1.8 V, with the leakage current for battery in HE less than that in AE, confirming better stability of Ti-foil in HE.

SEM images (Fig. S35d-f) of the Ti-foil from batteries after 300 cycles exhibit smooth surface(s) with no apparent corrosion.

Supplementary Fig. S35. Leakage current for Ti-foil in, (a) AE and (b) HE over time. (c) Stabilized leakage currents for Ti-foil at open-circuit voltage, 1.2, 1.5 and 1.8 V, respectively. SEM images of fresh Ti-foil (d) worked for 300 cycles in (e) AE and (f) HE.

Corrosion of Zn derives from side reactions of metallic Zn and electrolytes. In practical applications batteries are not always under continuous-work conditions, therefore, we established the degree corrosion for Zn in AE and HE in respectively, stand-by and work conditions.

As is seen in **Fig. S14b** significant corrosion occurs when fresh Zn-foil is immersed in AE for 72 h in which the Zn-foil is covered with a layer of corrosion products. FTIR spectra, **Fig. S14b**, confirm that this layer is derived from the decomposition of OTf⁻ anions. However, Zn-foil immersed in HE exhibits a smooth surface with only 'slight' corrosion spotting, **Fig. S14c**. Tafel plots for Zn-foil in AE and HE were determined to compare degree of corrosion in the two electrolytes. To obviate interference of Zn deposition, Zn²⁺ ion in AE and HE was replaced with Li⁺ ion. The corrosion current (i_{corr}) in AE was 39.06 $\mu\text{A cm}^{-2}$, whilst that in HE was reduced to just 7.07 $\mu\text{A cm}^{-2}$, confirming the corrosion-inhibiting effect of HE. A quantitative analysis of the corrosion rate of metallic Zn in these two electrolytes was conducted *via* monitoring potential changes for Zn in time. Initially, a fixed amount of Zn, 0.5 mAh, was deposited onto the Ti-foil to form a Zn@Ti electrode, and the Zn@Ti electrode exhibited the potential of metallic Zn. Once all metallic Zn was corroded by electrolyte, the potential for the Zn@Ti electrode increased significantly. **Fig. S15** evidences that the corrosion rate for Zn in AE is *ca.* 0.043 mg h⁻¹, whilst that in HE is highly significantly reduced to just 0.003 mg h⁻¹.

In work condition the interface of Zn and the electrolyte changes dynamically, and the corrosion products affect Zn deposition morphology. X-ray photoelectron spectroscopy (XPS) was used to analyse surface chemistry of Zn-foil that had been cycled 100 times in AE and HE. For the Zn-foil cycled in AE, strong signals appeared in the S 2p and F 1s spectrum, evidencing side reaction products from electrolyte decomposition on Zn-foil, specifically, ZnSO₄, ZnS and ZnF₂¹⁵, **Fig. 3c**. No apparent by-products however were detected on the Zn-foil cycled in HE, **Fig. 3d**, confirming the side reactions between electrolyte and metallic Zn are significantly suppressed. The significantly improved stability for HE is attributed to the strong DMAC/TMP-H₂O and DMAC-OTf⁻ interaction that restricts activity of water and anion. Moreover, a preferred orientation for the Zn (002) plane is found for the Zn-foil cycled 100 times in HE, confirming that the close-packed growth model with a layer-by-layer structure occurs in initial plating and is maintained in following cycling, **Fig. 3f**. In contrast, the Zn-foil cycled in AE has a 'pulverized' surface which may results in loss of active metallic Zn, **Fig. 3e**. Coulombic efficiency (CE) for Zn plating/stripping is a quantitative measurement for the reversibility of Zn anode and is affected by corrosion of Zn. **Fig. 1f** presents the CE for Zn||Cu batteries using AE and HE as electrolyte, respectively, at a current density 1 mA cm⁻² and plating/stripping capacity 1 mAh cm⁻². It is seen in the figure that Zn electrode in HE exhibits a highly significant improvement in stability and reversibility with a CE as high as 99.5 %.

Both in stand-by and work condition, OTf^- anions are involved in side reactions in AE. The solvation environment of for OTf^- anions was established *via* Raman spectroscopy, Fig. 3a. Findings show that the peaks, representing the $-\text{CF}_3$ symmetric and asymmetric deformation mode for $\text{Zn}(\text{OTf})_2$, shift slightly in AE, evidencing an interaction between H_2O and OTf^- anion. The signals for $-\text{CF}_3$ shift when $\text{Zn}(\text{OTf})_2$ is dissolved in TMP and DMAC, evidencing strongly that OTf^- anion coordinates with them. However, the signals for $-\text{CF}_3$ overlap with that for TMP and DMAC molecules (Supplementary Fig. S13), making it practically difficult to identify which solvent molecule the OTf^- anion interacts with mostly within the HE. Nuclear magnetic resonance (NMR) spectroscopy was used as complementary data to determine the chemical environment of the OTf^- anion. Fig. 2b shows the ^{19}F resonance frequencies for $\text{Zn}(\text{OTf})_2$ solutions with H_2O , TMP and DMAC as a single solvent, respectively. The ^{19}F signal for these electrolytes appears as a 'sharp', narrow peak at *ca.* -77.8 ppm. Because of different coordination molecules, the chemical shifts are different. The chemical shift for ^{19}F in HE is the same as that in DMAC, evidencing that OTf^- anions have the same chemical environment in DAMC solution and HE. In other words, OTf^- anions are surrounded mainly by DMAC molecules in HE. In a single solvent, DMAC, H_2O and TMP all coordinate with OTf^- anion to some degree, however, competition for anions occurs in DMAC/ H_2O /TMP mixture. DMAC molecule gains because its electron-rich acyl group exhibits much stronger interaction with $-\text{CF}_3$, a strong electron-withdrawing group. Consequently, the OTf^- anion in HE is similar in character to DMAC solution, with (almost) no anion decomposition.

Supplementary Fig. S14. SEM images of (a) fresh Zn-foil and these immersed in AE (b) and (c) HE for 72 h. (d) FTIR spectra for Zn-foil immersed in AE and HE. (e) Tafel curves and corresponding corrosion current density for Zn anode in AE and HE. To avoid interference of Zn deposition, Zn^{2+} ion in AE and HE is replaced with Li^+ ion.

Supplementary Fig. S15. Voltage-time curves for Zn@Ti electrodes immersed in AE and HE, respectively. An amount of metallic Zn, 0.5 mAh, was deposited on Ti-foil to form Zn@Ti electrode.

Fig. 1 (f) CE for Zn||Cu cells using AE and HE as electrolyte, respectively. Current density is 1 mA cm^{-2} and plating capacity is 1 mAh cm^{-2} .

Fig. 3. Analysis of side reactions between electrolyte and Zn anode. (a) Raman spectra for Zn(OTf)₂ and selected electrolytes. (b) ¹⁹F NMR spectra for selected electrolytes. High-resolution XPS spectra for Zn-foil following working in (c) AE and (d) HE for 100 cycles. (e-f) Morphology of Zn-foil following working in AE/HE for 100 cycles.

We have in our R-SI, Page 43 line 5-10, R-MS, Page 10 line 24-35 and Page 11 line 1-13, included the discussion as follow:

*“Because during operation of ZIBs, the applied voltage on Ti-foil can be up to 1.8 V, the corrosion of the Ti current collector was therefore determined via measuring leakage current during voltage hold. **Fig. S35a-c** evidences that the leakage current remains low for both AE and HE at a high voltage of 1.8 V, with the leakage current for battery in HE less than that in AE, confirming better stability of Ti-foil in HE. SEM images (**Fig. S35d-f**) of the Ti-foil from batteries after 300 cycles exhibit smooth surface(s) with no apparent corrosion.”*

*“Corrosion of the Zn anode is investigated by immersing the Zn foil in AE and HE for 72 hours. Zn foil suffered severe corrosion in AE and was covered with a layer of anion-derived byproducts (Supplementary **Figs. S14**). By contrast, the corrosion issue of the Zn anode is significantly mitigated in HE, which is also confirmed by the smaller corrosion current (*i*_{corr}) obtained in the Tafel plots (Supplementary **Figs. S14e**). A quantitative analysis of the corrosion rate of metallic Zn in these two electrolytes is conducted by monitoring the potential changes of Zn over time. At first, a fixed amount of Zn, 0.5 mAh, is deposited onto the Ti foil to form a Zn@Ti electrode, and once all the metallic Zn is corroded by the electrolyte, the potential of the Zn@Ti electrode soars. It is calculated that the corrosion rate of Zn in AE is around 0.043 mg h⁻¹, while that in HE is significantly reduced to only 0.003 mg h⁻¹, (Supplementary **Figs. S15**).”*

“X-ray photoelectron spectroscopy (XPS) was used to analyze the surface chemistry of Zn foil that had been cycled 100 times in various electrolytes. No anion-derived interphase components were observed on the Zn-foil cycled in electrolyte with DMAC or TMP as a single solvent (Supplementary Figs. S16 and S17). For the Zn foil cycled in AE, however, strong signals appeared in the S 2p and F 1s spectrum, evidencing side reaction products from electrolyte decomposition on Zn-foil, specifically, ZnSO₄, ZnS and ZnF₂¹⁵, Fig. 3c. In contrast, no apparent by-products were detected on the Zn-foil cycled in HE, Fig. 3d, confirming the side reactions between electrolyte and metallic Zn are significantly suppressed. The significantly improved stability of HE is attributed to the strong DMAC/TMP–H₂O and DMAC–OTf[−] interaction that restricts activity of water and anion. Moreover, a preferred orientation for the Zn (002) plane is found for the Zn-foil cycled 100 times in HE, confirming that the close-packed growth model with a layer-by-layer structure occurs in the initial plating and is maintained in the following cycling, Fig. 3f. In contrast, the Zn-foil cycled in AE has a ‘pulverized’ surface which may result in the loss of active metallic Zn, Fig. 3e.”

Comment 2-3

(c) Reading the response to the reviewers, I agree with the comments of reviewer three regarding the novelty of the materials tested. The authors' response is weak on this point, and it is a crucial weakness concerning the impact of this scientific article.

Response

We agree.

In response to address directly this comment we have reorganized the statement of novelty as follows:

In this work we propose a new strategy to reduce water reactivity and achieve dendrite-free Zn anode *via* introduction of strong polar molecules in aqueous electrolyte for Zinc ion batteries. The strong polar molecules, DMAC and TMP, interact with water, increase electron density of water protons and strengthen O–H bonds of water, thereby stabilizing water molecules and reducing side reactions with electrodes in the batteries. DMAC affects crystallographic orientation of Zn and induces exposure of the (002) plane if the ratio of DMAC and water is in a particular range leading to a dendrite-free Zn anode.

Organic solvents including, DMSO, DMC, DMA, TEP, NMP, EG, PC and DEC, have been introduced into aqueous electrolytes to suppress water reactivity *via* regulating solvation structures e.g. *J. Am. Chem. Soc.*, 2020, 142, 51, 21404, *Adv. Mater.*, 2019, 31, 1900668, *ACS Nano*, 2022, 16, 6, 9667 and *J. Am. Chem. Soc.*, 2022, 144, 16, 7160. The mechanism(s) is to break hydrogen bonds between water molecules, restrict activity of water in solvation sheaths, or exclude water molecules from the electric double layer (EDL), which aims to suppress water decomposition (reduction and oxidation). Although hybrid electrolytes with various components are reported, it hasn't been recognized that water reactivity can be suppressed *via* modulating the nuclear charge of the water protons. Here, we point out that by increasing the electron density of the water protons, electrostatic attraction between O–H will increase. DMAC and TMP are selected because they are strong polar molecules with electron-rich regions, C=O and P=O groups, that have interactions with water to increase electron density of the water protons, as is evidenced in DFT, NMR and FT-IR findings.

For Zn deposition, the (002) plane preferred orientation is an ideal growth pattern because it aligns the horizontal direction and therefore avoids dendrite formation. In this work we found that DMAC facilitates exposure of the (002) plane by altering the surface energy of the facets of Zn metal. This phenomenon had not been reported. The lone pair of electrons on the nitrogen atom makes DMAC a strongly polar molecule, which interacts with the metal surface *via* electrostatic interactions and therefore influences the surface energy of facets. The findings from density functional theory (DFT) computations evidence that Zn (002) exhibits the lowest surface energy in the mixture of DMAC/TMP/H₂O (5:2:3 by volume) compared with the (100) and (101) planes. This thermodynamic difference means that metallic Zn maximizes the expression of the (002) plane to minimize the total surface energy of the crystal, accounting for the dominant (002) plane in metallic Zn growth. This makes it possible to use a thin separator of 130 μm in the battery test instead of a 740 μm glass-fibre membrane. Importantly, this reduces mass and volume of the separator, together with reducing electrolyte used.

We have in our R-MS, Page 4 line 24-35, Page 5 line 1-3, line 6-9, Page 8 line 33-34, Page 9 line 1-11, included the discussion as follow:

“Various non-aqueous solvents, such as dimethyl sulfoxide (DMSO)²¹, dimethyl carbonate (DMC)²², N-methylpyrrolidone (NMP)²³, triethyl phosphate (TEP)²⁴, diethyl carbonate (DEC)²⁵, propylene carbonate (PC)²⁶ and ethylene glycol (EG)²⁷, were added into aqueous electrolytes to regulate the solvation structures²⁸. The mechanism(s) is to break hydrogen bonds between water molecules, restrict activity of water in solvation sheaths, or exclude water molecules from the electric double layer (EDL), which aims to suppress water decomposition. In this work, we proposed that water reactivity can be suppressed by increasing the electrostatic attraction between O–H via increasing the electron density of the water protons. Dimethylacetamide (DMAC) and trimethyl phosphate (TMP) are selected because they are strong polar molecules with electron-rich regions, C=O and P=O groups, that exert interaction on both solvated and free water molecules, and therefore impact the O–H bond strength of water. The decomposition of H₂O in the hybrid electrolyte is less thermodynamically favorable compared with that in aqueous electrolyte.”

“DMAC and TMP could alter the surface energy of Zn and therefore guide the (002) plane preferred orientation of Zn deposition, which boosts the lifespan of the Zn anode to > 1600 hours despite high applied current density and plating capacity of, respectively, 5 mA cm⁻² and 5 mAh cm⁻².”

*“Generally, crystallographic texture is determined by different growth rates amongst crystallites of various orientations, governed by surface energy minimization³⁴. Density functional theory (DFT) computations were therefore carried out to determine the surface energy of three main planes, namely, Zn (002), (100) and (101), in a solvent environment. Surface energy, the excess energy presented at a particular surface, determines the exposed facets and growth orientation of the crystal. It can be modified by solvent molecules via interactions^{35, 36}. As a result, the surface energy values for Zn (002), (100) and (101) alter when they are exposed to different solvents. Supplementary **Fig. S10** shows that Zn (002) exhibits the lowest surface energy of 1.33 J m⁻², in the mix of DMAC/TMP/H₂O (5:2:3 by volume) compared with the (100) and (101) planes, which are, respectively, 1.68 and 1.76 J m⁻². This thermodynamic difference implies that the metallic Zn maximizes expression of the (002) plane to minimize the total surface energy of the crystal, accounting for the dominant (002) plane in metallic Zn growth.”*

Comment 2-4

(d) Another remark concerns the quality of the figures. This is a formal weakness that the authors should improve because the article loses readability, especially for figure 5, which is too full of information and is in small type.

In conclusion, weaknesses b and c are too strong to accept the article, despite the technical quality of the results. The novelty of the work is an indispensable criterion for validation in such a journal as Nature Communication.

Response

Thank you for your suggestion. During the submission, all the documents in Word version were converted into PDF versions in the submission system. We noticed that the resolution of the figures was reduced greatly after conversion. In response to this comment, we have improved all figures for better readability. The high-resolution figures are also attached as supplementary documents for your kind reference.

Fig. 1. Electrolyte structure and water O-H bond characterization. (a) RDFs for Zn²⁺-O pairs in HE. (b) FTIR absorption and second derivative spectra for AE and HE. (c) ¹H NMR spectra for pure water, AE and HE. (d) Deprotonation (H⁺ dissociation of H₂O) energy from Zn²⁺ inner solvation shell of AE and HE. (e) ESW for AE and HE measured with 10 μm Pt ultramicroelectrode as working electrode at a scan rate 1 mV s⁻¹. (f) CE for Zn||Cu cells using AE and HE as electrolyte, respectively. Current density is 1 mA cm⁻² and plating capacity is 1 mAh cm⁻².

Fig. 2. Surface morphology for deposited Zn in (a) AE and (b) HE. (c) Optical microscopy image of Zn/electrolyte interface during Zn deposition in AE and HE. (d) Voltage profile for Zn||Zn symmetric battery tested at, respectively, high current density and high deposition capacity with AE and HE as electrolyte. (g) Surface energy value for main-planes of metallic Zn in liquid environment of DMAC/TMP/H₂O (5:2:3 by volume), H₂O and DMAC, respectively.

Fig. 3. Analyses of side reactions between electrolyte and Zn anode. (a) Raman spectra for $\text{Zn}(\text{OTf})_2$ and selected electrolytes. (b) ^{19}F NMR spectra for selected electrolytes. High-resolution XPS spectra for Zn-foil following working in (c) AE and (d) HE for 100 cycles. (e-f) Morphology for Zn-foil following working in AE/HE for 100 cycles.

Fig. 4. (a) Rate performance for Zn||NaV₃O₈ cells. (b) Energy and power density for Zn||NaV₃O₈ cells based on total mass of cathode and anode electrode. (c) Cyclic stability and CE for Zn||NaV₃O₈ cells at current density 0.5 A g^{-1} .

Fig. 5. Contour plot for NaV_3O_8 (204) reflection using (a) AE and (b) HE as electrolyte, in which the (204) reflection evolves together with corresponding charge/discharge curve. Synchrotron-based XRPD patterns for NaV_3O_8 electrodes following cycling in (c) AE and (d) HE.

Fig. 6. (a) 3-electrode CV curves for NaV_3O_8 electrode tested in HE. (b) Specific capacity of NaV_3O_8 electrode tested at temperature ranging from -40 to 70 °C. (c) Cyclic performance for $\text{Zn}||\text{NaV}_3\text{O}_8$ cells at 70 °C with AE and HE as electrolyte. (d) Cyclic performance at -40 °C in the presence of HE (HE is liquid at -40 °C).

Reviewer #3 Remarks to Author

The revised manuscript satisfies all comments of the reviewer, thus, I recommend the acceptance of this manuscript for publication.

Response

We thank Reviewer #3 for his/her recommendation.

END OF RESPONSE TO REVIEWS